# Dorsomorphin inhibits AMPK, upregulates *Wnt* and *Foxo* genes and promotes the activation of dormant follicles
Julie Feld Madsen[1], Emil Hagen Ernst[2,3], Mahboobeh Amoushahi[1], Margit Dueholm[2], Erik Ernst[1,4] & Karin Lykke-Hartmann [1,5] ✉

AMPK is a well-known energy sensor regulating cellular metabolism. Metabolic disorders such as obesity and diabetes are considered detrimental factors that reduce fecundity. Here, we show that pharmacologically induced in vitro activation (by metformin) or inhibition (by dorsomorphin) of the AMPK pathway inhibits or promotes activation of ovarian primordial follicles in cultured murine ovaries and human ovarian cortical chips. In mice, activation of primordial follicles in dorsomorphin in vitro-treated ovaries reduces AMPK activation and upregulates *Wnt* and *FOXO* genes, which, interestingly, is associated with decreased phosphorylation of β-catenin. The dorsomorphin-treated ovaries remain of high quality, with no detectable difference in reactive oxygen species production, apoptosis or mitochondrial cytochrome c oxidase activity, suggesting safe activation. Subsequent maturation of in vitro-treated follicles, using a 3D alginate cell culture system, results in mature metaphase eggs with protruding polar bodies. These findings demonstrate that the AMPK pathway can safely regulate primordial follicles by modulating *Wnt* and *FOXO* genes, and reduce β-catenin phosphorylation.

In the mammalian ovary, a finite number of nonrenewable follicles are assembled in the fetal ovaries and serve as the reservoir for future fertility. Fertility can be preserved for decades, as the majority of follicles are maintained dormant as primordial follicles[1,2]. A primordial follicle is a unit composed of an oocyte surrounded by pregranulosa cells[3]. In the cycling year, each month, a small fraction of primordial follicles is selectively activated, constituting the committing step into folliculogenesis, a process essential for the fertility outcome in women[1,4]. Recently, more biochemical pathways and molecules believed to maintain the balance between dormancy and activation of primordial follicles have been revealed predominantly from transgenic mouse models[5–7]. A well-known pathway implicated in this process is the phosphatidylinositol-3-kinase (PI3K)/ protein kinase B (AKT) pathway. The PI3K/AKT pathway plays a vital integrative role in linking many of the factors associated with the balance between follicle growth suppression, activation, and the maintenance of heathy quiescence[5,8]. Molecules in this pathway include phosphatase and tensin homolog (PTEN)[9], forkhead box O3 (FOXO3A)[10,11] and protein 27 (p27)[12], which maintain dormancy, whereas mammalian target of rapamycin (mTOR) acts as an activator[6,7,13,14]. Theoretically, the potential to manipulate the rate of activated primordial follicles from the ovarian reserve may have great potential in uncovering novel curative treatments for

infertility[15,16]. Women momentarily suffering from a low ovarian reserve and requiring assisted conception due to diverse aetiologies, including genetics, advanced reproductive age or iatrogenic causes, could benefit from increased primordial follicle activation to recover more oocytes[15,17]. Conversely, women facing gonadotoxic chemotherapy or an increased risk of early menopause may benefit from slowed activation to avoid premature ovarian insufficiency[5,18,19]. Global transcriptome analysis of gene expression patterns in oocytes and granulosa cells from primordial and primary follicles showed that the gene encoding 5'AMP-activated protein kinase (AMPK) was downregulated in oocytes from primordial to primary follicles[20,21]. AMPK is a highly conserved sensor of the cellular energy status. Once activated, AMPK switches off anabolic pathways and pathways consuming ATP while switching on catabolic pathways to restore cellular energy homeostasis[22,23]. In a conditional knockout mouse with an oocyte-specific deletion of the *Lkb1* (liver kinase B1) gene, an upstream kinase of AMPK, the entire primordial follicle pool was activated but failed to mature and ovulate, resulting in premature ovarian failure in early adulthood[24]. In line with these findings, in vitro culture studies of murine ovaries with dorsomorphin (an inhibitor of AMPK) followed by in vivo grafting under separate sides of the kidney capsule in ovariectomized hosts were shown to stimulate preantral follicle growth[25]. A previous study indicated that metformin (activator of

[1]Department of Biomedicine, Aarhus University, DK-8000 Aarhus C, Denmark. [2]Department of Obstetrics and Gynaecology, Aarhus University Hospital, DK-8000 Aarhus C, Denmark. [3]Department of Gynaecology and Obstetrics, Gødstrup Hospital, DK-7400 Herning, Denmark. [4]Fertility Clinic Regional Hospital Horsens, DK-8700 Horsens, Denmark. [5]Department of Clinical Genetics, Aarhus University Hospital, DK-8200 Aarhus N, Denmark. ✉e-mail: kly@biomed.au.dk

AMPK) is a novel promising option for preserving ovarian function and fertility during chemotherapy[26]. Despite the association of AMPK with follicle development, it is unknown how this might be beneficial in terms of clinical perspectives in infertility treatments and how this affects intracellular signaling in the oocyte. Additionally, there are no data on the safety of the oocyte using AMPK as a target for regulating primordial follicle growth.

Here, we demonstrate the collaborative role of the AMPK pathway in conjunction with the PI3K/AKT pathway in influencing the rate of primordial follicle activation, ultimately resulting in the production of high-quality oocytes. Notably, the inhibition of AMPK leads to an upregulation of *Foxo* and *Wnt* genes. Collectively, our findings show the regulatory contributions made by AMPK regulation. The ability to promote activation of primordial follicles holds promise to increase the likelihood of obtaining more mature eggs, dependent on several aspects, including safety, signaling, diseases and age. RNA sequencing data of the dorsomorphin-induced oocytes from primordial follicles revealed the action to be linked to modulating *Wnt* and *FOXO* genes, thereby reducing β-catenin phosphorylation. It is important to emphasize that the endogenous activation of primordial follicles involves a complex network of various factors. This network encompasses signaling mechanisms within individual follicles, and between adjacent follicles, and signaling from the hypothalamic-pituitary-ovarian axis, which collectively governs female reproduction

## Results
### *PRKAA1* and *PRKAA2* transcripts and proteins are expressed in primordial and developing follicles in humans and mice

AMPK is composed of a catalytic α and two regulatory β and γ subunits[22]. In mammals, the catalytic subunit is encoded by two alternate genes, the *PRKAA1* and *PRKAA2* genes[22]. The *PRKAA1* and *PRKAA2* gene expression profiles imply a role in follicle activation, as the *PRKAA1/2* genes were noted

to be highly expressed in human oocytes from primordial follicles and less expressed in oocytes from primary follicles (Fig. 1a and Supplementary Fig. S1 for FPKM values for oocytes and granulosa cells). This shows that the *PRKAA1* and *PRKAA2* genes are expressed in dormant human primordial follicles at high levels, with differential expression as the follicles are activated and become primary. When AMPK is activated, it phosphorylates and regulates downstream targets, e.g., genes such as *PPARGC1A* (peroxisome proliferator-activated receptor gamma coactivator 1-alpha) and *ACACA* (acetyl-CoA carboxylase), promoting energy conservation and production while inhibiting anabolic processes. Both *PPARGC1A* and *ACACA* were shown to be highly expressed in the oocytes of primordial and primary follicles, highlighting AMPK's involvement in primordial follicle activation (Fig. 1a) (Supplementary Table 1).

In line with the mRNA transcripts, the alpha1 and alpha2 isoforms of the AMPK protein were found to be expressed in both primordial and primary follicles both in humans (Fig. 1b) and in mice (Fig. 1c), indicating a role of AMPK in the primordial to primary follicle transition (Supplementary Fig. S1 for negative controls and antibody validation).

### AMPK inhibition promotes the activation of mouse and human dormant (primordial) follicles

To explore the potential regulatory role of AMPK in primordial follicles, metformin and dorsomorphin were used to antagonize AMPK function. Ovaries were removed from juvenile mice and cultured in vitro with medium supplemented with metformin or dorsomorphin (Fig. 2a). After metformin and dorsomorphin in vitro treatments, a stereological assessment of cultured ovaries was performed, counting primordial, primary and secondary follicle stages. Primordial follicles were morphologically distinguished as an oocyte encapsulated by flattened granulosa cells (Fig. 2b, PrF), while the oocytes in primary follicles were thus surrounded by one

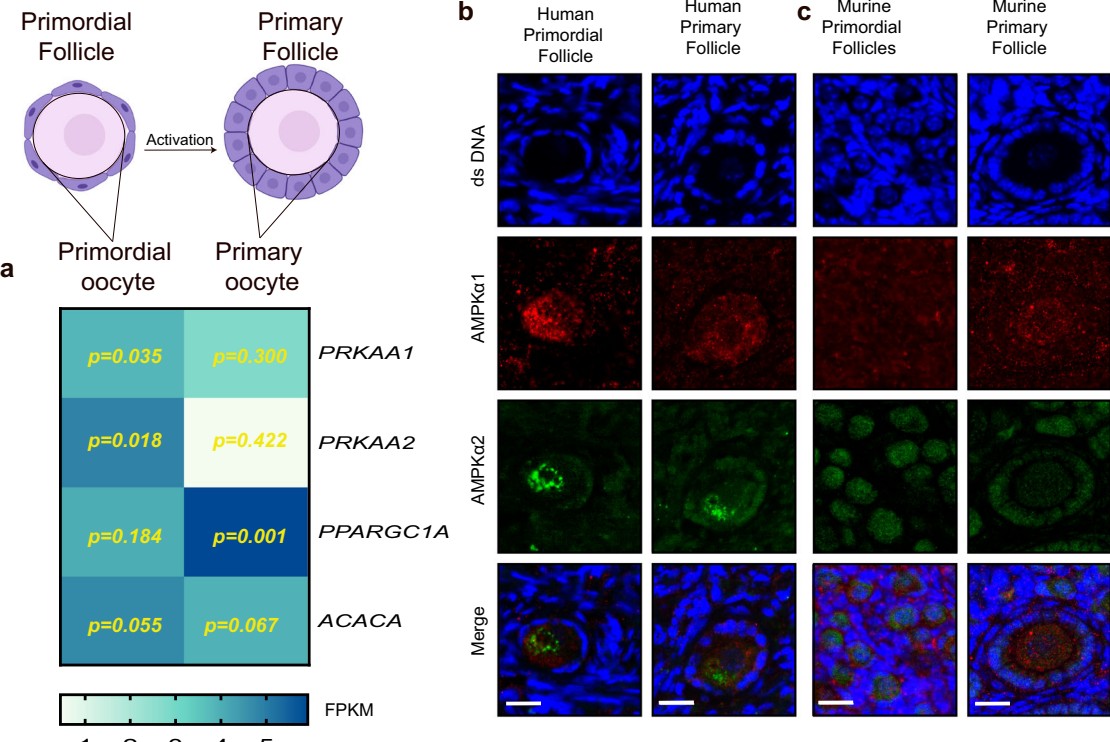

**Fig. 1 | Expression of PRKAA1 and PRKAA2 mRNA and protein in early-stage follicles in humans and mice. a** Schematic illustration of primordial and primary follicles. The dormant primordial follicle is composed of an oocyte (pink) surrounded by flattened pregranulosa cells (purple). Upon follicle activation, the follicle becomes primary, and the oocytes are now surrounded by cuboidal granulosa cells. Heatmap of *PRKAA1*, *PRKAA2*, *PPARGC1A*, and *ACACA* gene expression (FPKM values) in human oocytes from primordial and primary follicles based on RNA sequencing data. *P* values from consistent stage-specific expressed genes are noted. **b, c** Fluorescence microscopy images showing the distribution of AMPKα1 and AMPKα2 subunits in different follicular stages in human tissue. **b** and murine tissue (**c**), and DAPI staining shows double-stranded DNA (Ds DNA). Scale bar: 20 μm. **a** created with BioRender.com.

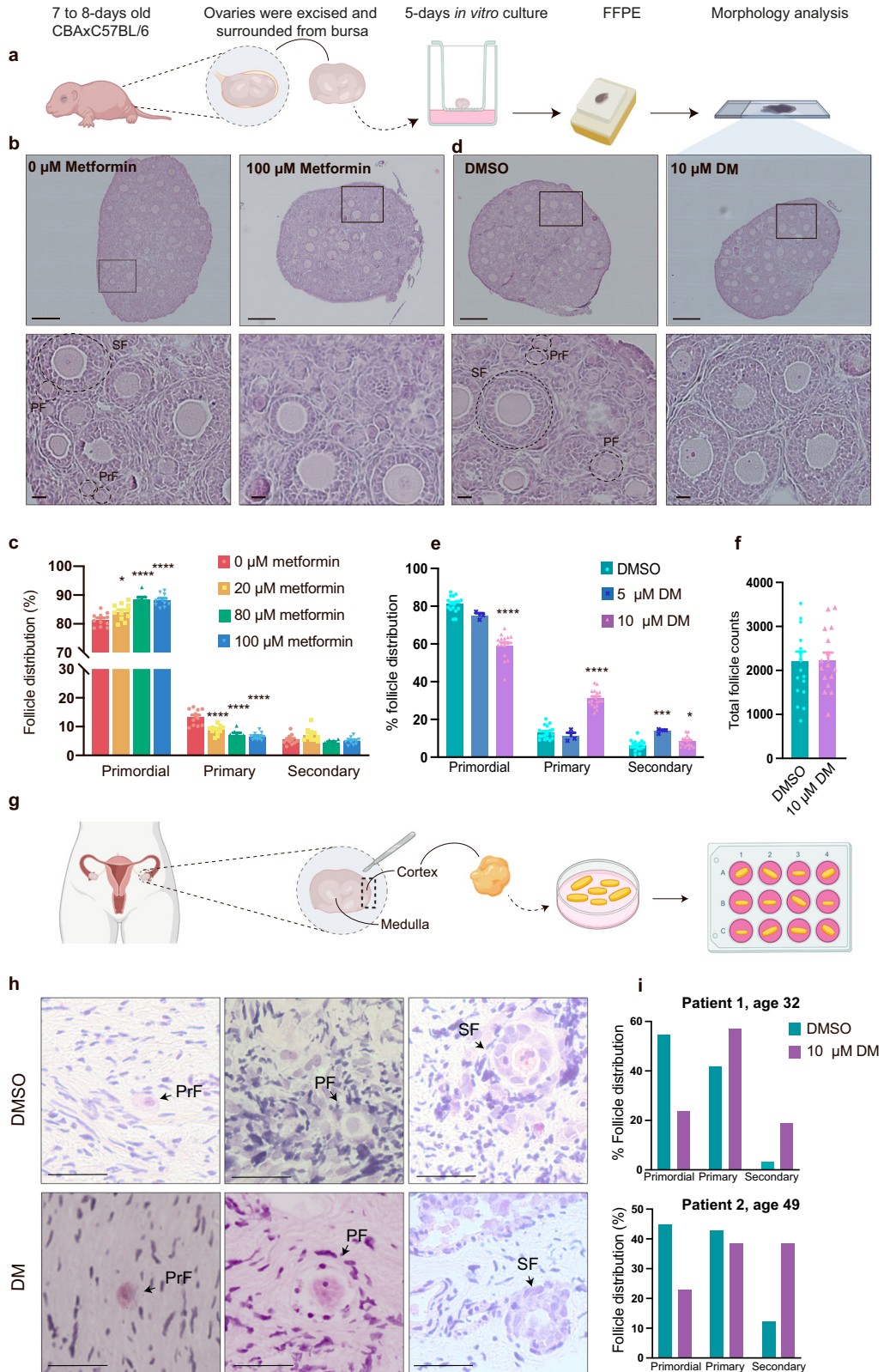

layer of cuboidal granulosa cells (Fig. 2b, PF), and secondary follicles had a minimum of two layers of granulosa cells around the oocyte (Fig. 2b, SF). Representative images of whole ovarian sections (Fig. 2b) and representative images of follicles at high magnification showed healthy-looking follicles, defined as containing an intact oocyte, organized granulosa cell layers, and no pyknotic bodies. In vitro culture of ovaries in medium supplemented

with metformin (Fig. 2b and Supplementary Fig. S2a–d) retained significantly more follicles in the dormant stage compared to control-treated ovaries in a concentration-dependent manner (Fig. 2c, Table 1). In contrast, in vitro culture of ovaries with medium supplemented with dorsomorphin (Fig. 2d and Supplementary Fig. S2e–g) significantly decreased the number of primordial follicles compared to the control (Fig. 2e, Table 2), indicating

**Fig. 2 | AMPK activation and inhibition can regulate dormancy and activation of the primordial follicle pool. a** Schematic illustration of the experimental design of primary in vitro ovary culture. **b** Representative photomicrographs of whole ovary sections and cortical sections of in vitro cultured ovaries treated with 0 or 100 μM metformin stained with H&E at high magnification showing decreased primordial follicle activation and follicle morphology, respectively. Identification of follicular stages. PrF primordial follicle, PF primary follicle, and SF secondary follicle. Scale bar: 20 and 200 μm. S2 for low magnification photomicrographs. **c** Follicle count of in vitro cultured ovaries supplemented with 0–100 μM metformin as a percentage. 0 μM metformin n = 11 biologically independent samples (ovaries), 20 μM metformin n = 11 biologically independent samples (ovaries), 80 μM metformin n = 7 biologically independent samples (ovaries), 100 μM metformin n = 11 biologically independent samples (ovaries). **d** Representative photomicrographs of whole ovaries and cortical sections of in vitro cultured ovaries treated with DMSO or 10 μM dorsomorphin stained with H&E at high magnification. Scale bar: 20 μm. S2 for low magnification photomicrographs. **e** Follicle count of in vitro cultured ovaries

supplemented with 0–10 μM dorsomorphin as a percentage. DMSO n = 17 biologically independent samples (ovaries), 5 μM DM *n = 3* biologically independent samples (ovaries), 10 μM DM n = 16 biologically independent samples (ovaries). **f** Total follicle count of ovaries exposed to DMSO n = 17 biologically independent samples (ovaries) or 10 μM DM n = 16 biologically independent samples (ovaries). **g** Schematic illustration of in vitro culture of human cortical ovarian pieces. **h** Histological H&E-stained sections representing a primordial, primary and secondary follicle after in vitro culture for two weeks with DMSO or dorsomorphin. Scale bar: 20 μm. **i** Follicle count of the in vitro cultured human ovarian cortical cubes, n = 2 biologically independent samples (ovary biopsy), the data are represented in percentage from total counts (Table 3). All data are represented as the mean ± SEM. In **c** and **e**, a one-way ANOVA was conducted, followed by multiple comparisons of the mean of each group to the mean of the control group. **f** An unpaired *t* test was conducted. *P < 0.05, **P < 0.01, ***P < 0.001, ****P < 0.000. **a** and **g** created with BioRender.com.

**Table 1 | Follicle distribution in ovaries cultured in vitro with metformin (0–100 μm)**

| Metformin (μM) | 0 μM | 20 μM | 80 μM | 100 μM | *p* value |
|---|---|---|---|---|---|
| Primordial follicle (%) | 81.36 ± 0.69 | 84.10 ± 0.84[(*)] | 88.38 ± 0.10[(****)] | 88.22 ± 0.63[(****)] | <0.0001 |
| Primary follicle (%) | 13.20 ± 0.84 | 8.83 ± 0.47[(****)] | 7.16 ± 0.70[(****)] | 6.55 ± 0.37[(****)] | <0.0001 |
| Secondary follicle (%) | 5.43 ± 0.52 | 7.07 ± 0.65 | 4.46 ± 0.37 | 5.03 ± 0.36 | 0.0082 |

0 μM n = 11, 20 μM n = 7, 20 μM n = 11, 100 μM n = 11. One-way ANOVA was applied to analyze the differences between the control and treatment groups, and multiple comparisons between the mean of the control group and the mean of each treatment group were made according to the Bonferroni correction.

**Table 2 | Follicle distribution in ovaries cultured in vitro with dorsomorphin (0–10 μM)**

| Dorsomorphin (μM) | 0 μM | 5 μM | 10 μM | *p* value |
|---|---|---|---|---|
| Primordial follicle (%) | 81.07 ± 1.02 | 75.03 ± 1.24 ns | 59.09 ± 1.69[(****)] | <0.0001 |
| Primary follicle (%) | 12.94 ± 0.83 | 11.10 ± 1.83 ns | 31.08 ± 1.16[(****)] | <0.0001 |
| Secondary follicle (%) | 6.02 ± 0.68 | 13.80 ± 0.72[*] | 8.37 ± 0.71[(***)] | 0.0002 |

0 μM n = 17, 5 μM n = 3, 10 μM n = 16. One-way ANOVA was applied to analyze the differences between the control and treatment groups, and multiple comparisons between the mean of the control and the mean of each treatment group were made according to the Bonferroni correction.

accelerated activation of primordial follicles. Notably, we checked the total follicle count (primordial, primary, secondary) once exposed to 10 μM dorsomorphin or DMSO for all ovaries (n = 16/17), which showed that ovaries cultured with DMSO contained 2207 ± 222.7 follicles, compared to the dorsomorphin-exposed ovaries, which contained 2229 ± 175.0 follicles (Fig. 2f). An unpaired t test showed that the differences were not significant (p = 0.9391) and thus it can be concluded that dorsomorphin does not change the number of total follicles but rather induces the activation of primordial follicles into primary or secondary follicles. The augmentation of primordial follicle activation with the AMPK inhibitor dorsomorphin was further substantiated through orthogonal validation with another AMPK inhibitor, BAY-3827. Findings indicated the ovaries exposed to BAY-3827 harbored significantly fewer primordial follicles (DMSO vs BAY-3827; 80.114 ± 5.001 *vs.* 66.157 ± 6.389, p = 0.0018, n = 6). In alignment with this, an increased count of primary follicles was observed following exposure to BAY-3827 (DMSO *vs* BAY-3827; 14.609 ± 5.526 *vs.* 27.925 ± 4.287, p = 0.0009, n = 6) (Supplementary Fig. S3).

We next investigated whether dorsomorphin could activate primordial follicles in human tissue obtained from two patients, aged 32 and 49 years old, with distinct hormone profiles (Fig. 2g) (Table 3). After two weeks of culture with or without dorsomorphin (Fig. 2h), the number of human primordial follicles decreased dramatically, while primary and secondary follicles had increased numbers in dorsomorphin-exposed ovarian fragments compared to DMSO-exposed fragments (Table 3), presented as percentages difference (Fig. 2i) clearly indicate that primordial follicles were activated and developed into primary and secondary follicles in human ovarian tissue.

**Table 3 | Hormone levels in patients donating the tissue based on serum measures and their corresponding follicle distribution of primordial, primary, and secondary follicles from ex vivo culture, with or without DM**

| | Patient 1 | Patient 2 |
|---|---|---|
| Age (years) | 32 | 49 |
| FSH[a] (int.enh/L) | 4.5 | 2.3 |
| LH[a] (int.enh/L) | 6.9 | 1.5 |
| AMH[a] (pmol/L) | 51.6 | <0.5 |
| p-oestradiol (pmol/L) | 326 | - |
| p-progesterone (nmol/L) | <0.7 | - |
| p testosterone (nmol/L) | 0.98 | - |
| Primordial follicles DMSO vs. DM | 17 vs. 5 | 22 vs. 3 |
| Primary follicles DMSO vs. DM | 13 vs. 12 | 21 vs. 5 |
| Secondary follicles DMSO vs. DM | 1 vs. 4 | 6 vs. 5 |

[a]Follicle stimulating hormone (FSH), luteinizing hormone (LH), and anti-Mullerian hormone (AMH).

### Dorsomorphin exposure suppresses AMPK activation independently of the PTEN/AKT signaling pathway

We next investigated how dorsomorphin causes activation of primordial follicles by examining whether this was caused by an inhibition of the AMPK pathway and independent of the classical PTEN/PI3K/AKT pathway known to play a central role in controlling the dormancy and activation

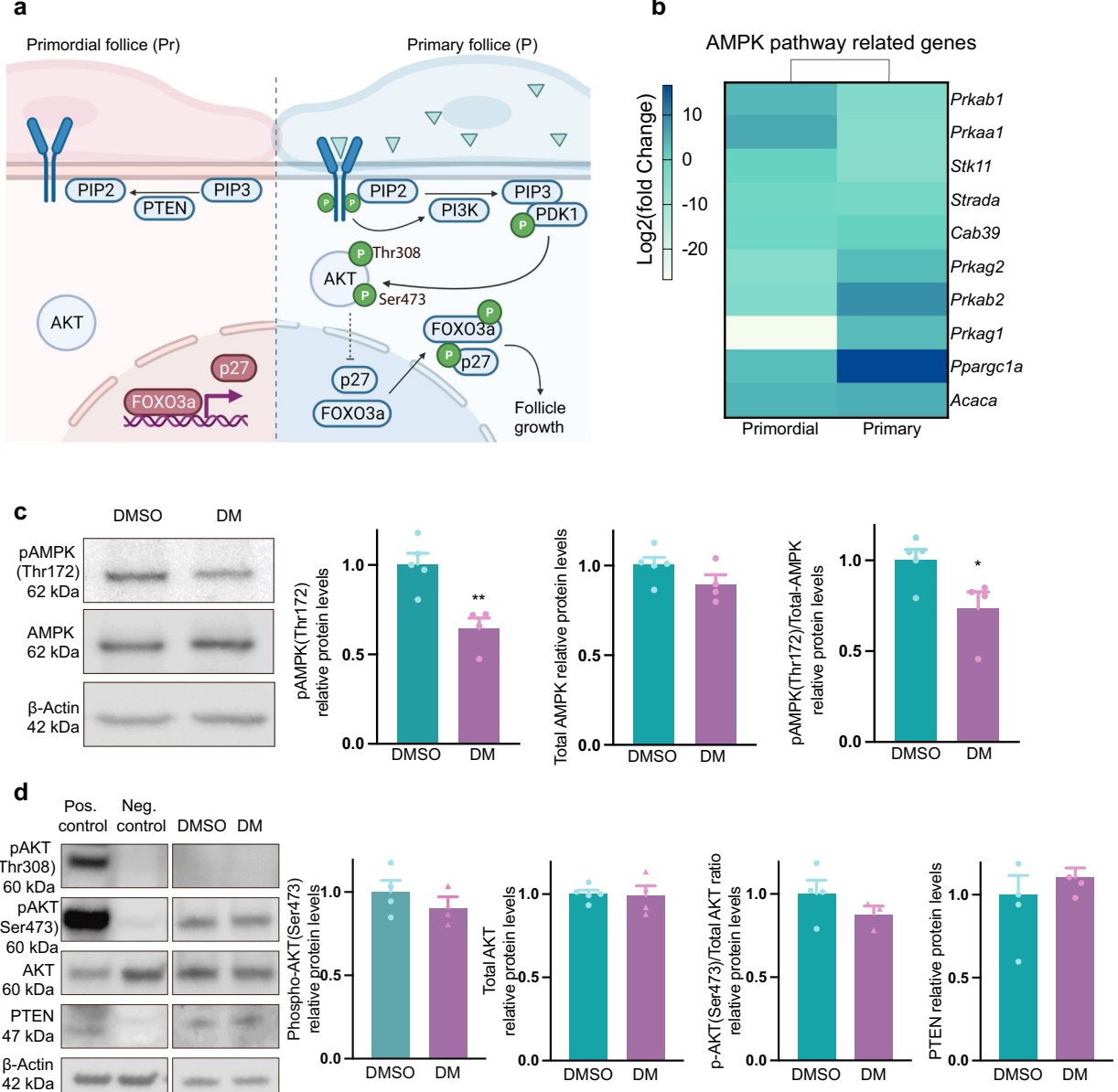

**Fig. 3 | Dorsomorphin exposure suppresses AMPK activation independently of the PTEN/AKT signaling pathway. a** Simplified diagram of the PI3K/AKT pathway in dormant and activated follicles. **b** Heatmap with log2fold change values of AMPK-related genes in murine oocytes from primordial $n = 3$ biologically independent LCM samples consisting of 200 pooled oocytes from primordial follicles/group and primary $n = 3$ biologically independent LCM samples consisting of 200 pooled oocytes from primary follicles/group, follicles exposed to dorsomorphin compared to DMSO (control). **c** Phospho (Thr172) AMPK and total AMPK protein abundance of whole lysates from ovaries exposed to DMSO $n = 4–5$ biologically independent samples (ovaries) consisting of 6 pooled ovaries/group or DM $n = 4, 5$ biologically independent samples (ovaries) consisting of 6 pooled ovaries/group for 6 h analyzed by western blotting, using β-actin as a loading control. $n = 24–30$. **d** Phosphorylated (Thr308) AKT, phosphorylated (Ser473) AKT, total AKT, and PTEN protein abundance in whole lysates from ovaries exposed to DMSO $n = 4–5$ biologically independent samples (ovaries) consisting of 6 pooled ovaries/group or DM $n = 4–5$ biologically independent samples (ovaries) consisting of six pooled ovaries/group for 6 h was analyzed by western blotting, using β-actin as a loading control. All data are presented as the mean ± SEM and were analyzed with an unpaired $t$ test. An example blot is shown; original full-length blots are provided in S4. *$P < 0.05$. a created with BioRender.com.

of primordial follicles (Fig. 3a). First, to obtain a global view of the consequences of the dorsomorphin-induced inhibition of AMPK, we performed RNA sequencing. We interrogated the transcriptomes of oocytes laser-isolated from primordial (GEO link: https://www.ncbi.nlm.nih.gov/geo/query/acc.cgi?acc=GSE230258.) and primary (GEO link: https://www.ncbi.nlm.nih.gov/geo/query/acc.cgi?acc=GSE230258) follicles, with and without dorsomorphin treatment, and performed a fold change of gene expression in each follicle stage, comparing the control to the dorsomorphin-treated group (GEO link: https://www.ncbi.nlm.nih.gov/geo/query/acc.cgi?acc=GSE230258). RNA sequencing analysis revealed

1664 differentially expressed genes (DEGs) (389 DEGs downregulated in the dorsomorphin group, 256 DEGs upregulated in the dorsomorphin group in oocytes from primordial follicles, and 308 DEGs downregulated in the dorsomorphin group, 711 DEGs upregulated in the dorsomorphin group in oocytes from primary follicles). We interrogated the transcriptomes of genes in the AMPK signaling pathway (Fig. 3b) (Supplementary Table 1) and found that the *Prkaa1* gene was highly downregulated. Similarly, the gene encoding the beta-1 subunit of AMPK, the *Prkab1* gene, was highly downregulated, confirming the effect of dorsomorphin. The gene encoding Strada, a Stik11 activating protein, was

likewise downregulated. The *Prkag1* and *Prkag2* genes are members of the AMPK gamma subunit family, and the *Prkab2* gene is upregulated. Furthermore, we assessed AMPK downstream substrate transcripts such as *Ppargc1a* and *Acaca*. *Acaca* was equally expressed in the oocytes from primordial and primary follicles, whereas *Ppargc1a* was upregulated in the oocyte of the primary follicle. However, in general, AMPK-related genes were downregulated, confirming the effect of dorsomorphin. To confirm this, we measured the protein levels of phosphorylated (P)-AMPK threonine (Thr) 172 compared to total AMPK levels after 6 h of in vitro culture with or without dorsomorphin. After culture, the levels of (P)-AMPK-Thr172/total AMPK were significantly ($p = 0.0355$) lower in ovaries exposed to dorsomorphin (Fig. 3c and Supplementary Fig. S4). In comparison, we determined whether and to what extent the PTEN/AKT signaling pathway contributes to the enhanced activation of primordial follicles observed in dorsomorphin-treated ovaries. To that end, we compared the levels of PTEN and levels of phosphorylated (P)-AKT compared to the total levels of AKT at the two different phosphorylation sites (Thr308 and Ser473). The results showed no alterations in PTEN levels ($p = 0.4863$) or the phosphor (Ser473)-AKT/total AKT levels ($p = 0.2886$), and both the control and dorsomorphin-exposure groups expressed a limited amount of phosphorylated AKT at Thr308, and levels could not be quantified. We suspect no differences in phosphor-(Thr308)-AKT levels (Fig. 3d and Supplementary Fig. S4). Considering the above observations, the enhanced number of activated follicles in dorsomorphin-exposed ovaries appears to be caused by a decrease in activated AMPK signaling, independent of the PTEN/AKT signaling pathway within the primordial follicle pool.

### Dorsomorphin exposure upregulates *Wnt* and *Foxo* genes but does not compromise nuclear exclusion of FOXO3A in activated follicles

Next, we wished to decipher how dorsomorphin supports the activation and growth of follicles. Insights from the RNA data revealed that wingless-type mouse mammary tumor integration site (*Wnt*) genes, in general, were upregulated in both primordial and primary follicles post dorsomorphin exposure, especially the *Wnt2b* gene, which was highly upregulated, and *Wnt4*, *Wnt5a* and *Wnt11* and to a lesser degree (Fig. 4a) (Supplementary Table 1). Wnt signaling is functionally linked to cellular metabolism[27], and it has been shown that AMPK inhibition by Compound C (dorsomorphin) reduced the phosphorylation of β-catenin[28], leading to the expression of canonical Wnt signaling. To test whether this pathway was active in ovaries, we investigated whether dorsomorphin exposure reduced the protein levels of phosphorylated β-catenin at serin 552 and found that dorsomorphin significantly reduced the levels of phosphorylated β-catenin at serin 552 ($p = 0.0133$) (Fig. 4b and Supplementary Fig. S6), confirming Wnt/β-catenin signaling in follicular development. As Wnt/β-catenin signaling regulates follicular development by modulating the expression of FOXO3A signaling[29], we noted that the transcript expression of both *Foxo1* and *Foxo3a* is upregulated in oocytes from primordial and primary follicles after dorsomorphin treatment (Fig. 4a).

Both the AMPK and PI3K/AKT signaling pathways integrate their biological information into FOXO3A[30]; likewise, p27 is also utilized by both pathways to integrate opposite signals[31,32], and both are crucial for maintaining good-quality oocytes[5–7,10,12]. We used immunofluorescence to determine the intracellular localization of FOXO3A and p27, as follicles activate and initiate growth. In ovaries exposed to dorsomorphin for five days, FOXO3A was expressed in the nuclei and cytoplasm of the oocytes of primordial and primary follicles (Fig. 4c). p27 was expressed in the nuclei of the oocytes of primordial and primary follicles but also in the cytoplasmic compartment as the follicle started to grow (Fig. 4d). In secondary follicles, the expression of p27 decreased in the oocyte; however, the expression of p27 was detected in the surrounding granulosa cells (Fig. 4d). A similar expression profile of FOXO3A and p27 was noted in control-treated ovaries (Supplementary Fig. S5 and negative controls). To further examine the translocation of FOXO3A and p27, we used Western blotting to quantify the translocation of the proteins and their negative regulator PTEN. Protein

levels of cytoplasmic levels of FOXO3A ($p = 0.4495$) and p27 ($p = 0.0128$) confirmed that nuclear exclusion of the proteins was not suppressed during follicular activation (Fig. 4e and Supplementary Fig. S6) (nuclear fractions see Supplementary Fig. S5). These results are supported by a lower quantification of the level of cytoplasmic PTEN ($p = 0.0737$) (Fig. 4f and Supplementary Fig. S7). The results indicate that dorsomorphin upregulated *Wnt* and *Foxo* genes when activating primordial follicles; however, it did not affect the nuclear exclusion of FOXO3A or p27, a downstream consequence of PI3K-AKT signaling activation.

### High-quality oocytes after dorsomorphin treatment

We next examined whether dorsomorphin affected the quality of the oocytes. We first explored the level of apoptosis by conducting a TUNEL assay (Fig. 5a, Supplementary Fig. S7 negative control). We detected a slight increase in the number of apoptotic cells in control ovaries; however, the result was not significant (DMSO: 6.43±0.838% *vs.* DM: 4.07±1.205%, $p = 0.8056$) (Fig. 5b). In line with this, we detected that the BAX/BCL-2 protein ratio was lower in ovaries exposed to dorsomorphin ($p = 0.0067$) (Supplementary Fig. S7). As a next step, we assessed mitochondrial function in ovaries exposed to dorsomorphin compared to DMSO by measuring cytochrome c oxidase activity and ROS production, as mitochondrial function plays an important role in the maturation of oocytes[33–35]. We found no significant differences in cytochrome c oxidase activity between DMSO and dorsomorphin treatment (DMSO: 1.476±0.462 units/mL *vs.* DM: 1.318±0.383 units/mL, $p = 0.8056$) (Fig. 5c). Nor did we observe any significant difference in the generation of ROS between the ovaries exposed to DMSO or dorsomorphin (DMSO: 283.1±23.25 vs. DM: 318.6±30.06 fluorescence intensity ($E_x/E_m = 488/525$ nm), $p = 0.4031$) (Fig. 5d). Interestingly, in addition to its cytoplasmic role in the regulation of the PI3K pathway, PTEN has also been proposed to localize to the nucleus and act crucially in maintaining genomic integrity[36–39]. We next examined whether dorsomorphin exposure impaired the balance between the cytoplasmic and nuclear abundance of PTEN using Western blotting. We observed that dorsomorphin exposure did not alter the cytoplasmic/nuclear ratio of PTEN ($p = 0.9416$) (Fig. 5e). Next, we interrogated the RNA sequencing data for genes related to oocyte quality. Of relevance to oocyte steroid metabolism, we noted no change in the *Cyp19a1* gene, encoding the aromatase enzyme (converting androgen to oestradiol), or in genes encoding proteins involved in steroidogenesis (e.g., *StaR*, *Cyp11a1*, and *Cyp17*) (GEO link: https://www.ncbi.nlm.nih.gov/geo/query/acc.cgi?acc=GSE230258.). We further interrogated the presence of genes known to be involved in oocyte competence in mice, and as expected, genes known to be involved in this process, e.g., the *murine double minute 2* (*Mdm2*)[40] and *Sirtuin 1* (*Sirt1*)[41–43] genes, were not detected. Taken together, our results suggest that dorsomorphin-induced primordial follicle activation does not compromise oocyte quality at any of the selected parameters.

### Oocytes from dorsomorphin-treated follicles resume meiosis and form metaphase II oocytes

The ultimate proof to test the quality of dorsomorphin-activated oocytes was to examine their follicular development and meiotic competencies. Therefore, we performed a long-term culture of ovaries. Ovaries were initially cultured for seven days in vitro supplemented with DMSO or dorsomorphin, and on day seven, secondary follicles were manually isolated and submitted to an additional 12-day culture in a three-dimensional culture system, where follicles then developed into antral follicles in medium containing neither dorsomorphin nor DMSO. During this long culture time, the number of follicles that survived, degenerated, formed and had an antrum were distinguished. Next, only follicles containing an antrum were selected for isolation of cumulus-oocyte complex (COC) and cultured overnight in a culture medium supplemented with an ovulating dose of human chorion gonadotropin (hCG). The extrusion of the first polar body was assessed the following day (Fig. 6a).

During the 12-day culture (Fig. 6b), we showed that dorsomorphin-activated follicles grew into antral follicles without jeopardizing the follicular

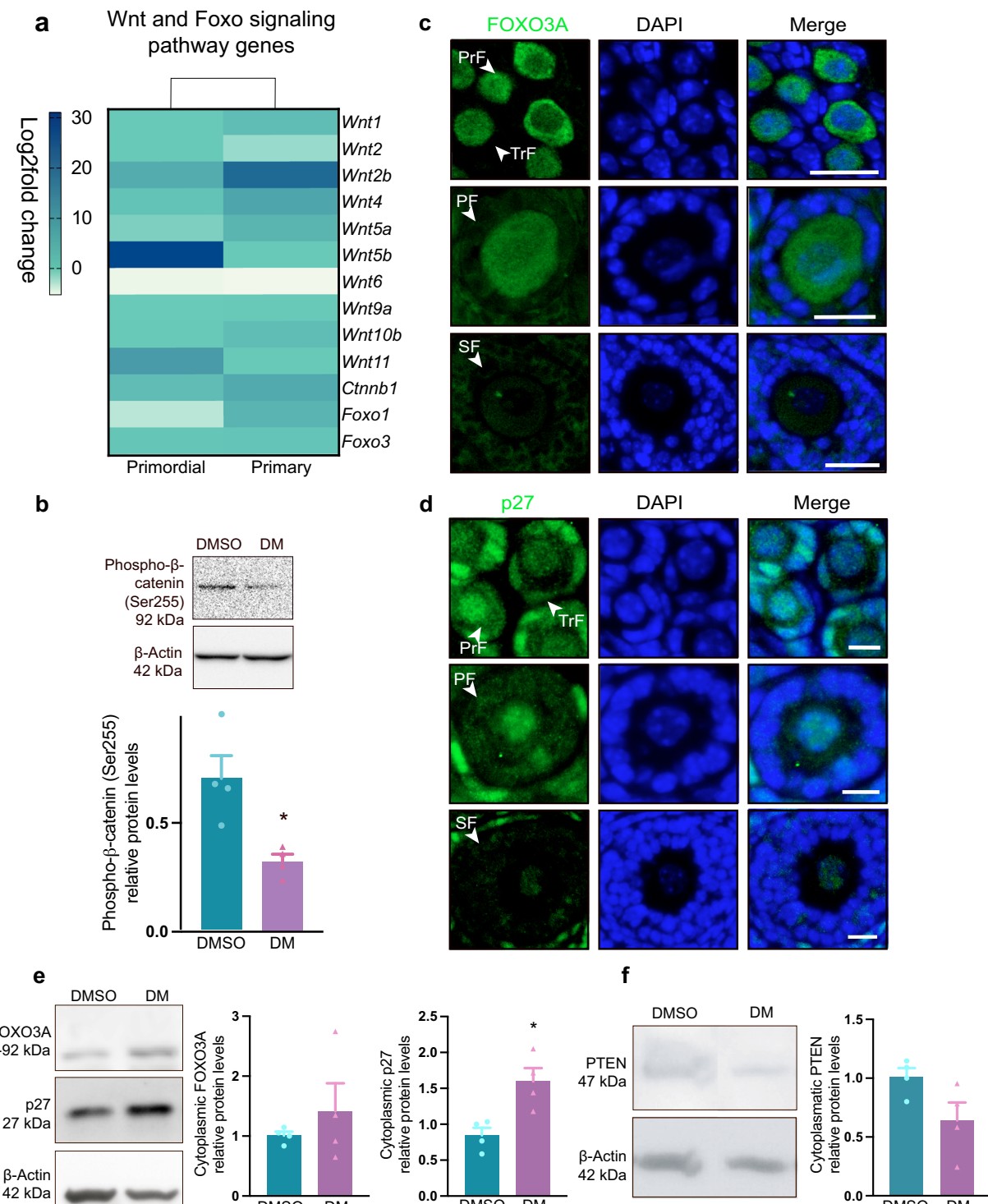

**Fig. 4 | Dorsomorphin exposure upregulates *Wnt* and *Foxo* genes but does not compromise nuclear exclusion of FOXO3A. a** Heatmap with Log2fold change values of *Wnt* and *Foxo* genes in primordial *n = 3* biologically independent LCM samples consisting of 200 pooled oocytes from primordial follicles/group and primary *n = 3* biologically independent LCM samples consisting of 200 pooled oocytes from primary follicles/group, follicles treated with dorsomorphin compared to control-exposure (DMSO). **b** Phospho-β-catenin (Ser255) protein abundance of whole lysates from ovaries exposed to DMSO *n = 4* biologically independent samples (ovaries) consisting of 4 pooled ovaries/group or DM *n = 4* biologically independent samples (ovaries) consisting of 4 pooled ovaries/group for 6 h analyzed by Western blotting, using β-actin as a loading control. *n* = 16. **c** Immunofluorescence staining was used to determine the localization of FOXO3A in primordial (PrF), transition (TrF), primary (PF), and secondary (SF) follicles in ovaries cultured in vitro with

dorsomorphin for five days. Scale bar: 20 μm. **d** Immunofluorescence staining was used to determine the localization of p27 in different follicular stages in ovaries cultured in vitro with dorsomorphin for 5 days. Scale bar: 20 μm. Immunofluorescence staining of control-treated ovaries is shown in Supplementary Fig. S5. **e** Detection of FOXO3A and p27 protein abundance in the cytoplasmic compartments normalized to β-actin. *n = 4* biologically independent samples (ovaries) consisting of 10–12 pooled ovaries/group. **f** Detection of PTEN protein abundance in the cytoplasmic compartment of ovaries cultured for five days using β-actin as a loading control. *n = 4* biologically independent samples (ovaries) consisting of 10–12 pooled ovaries/group. All data are presented as the mean ± SEM and were analyzed with an unpaired *t* test. An example blot is shown; original full-length blots are provided in Supplementary Figs. S4 and S6.

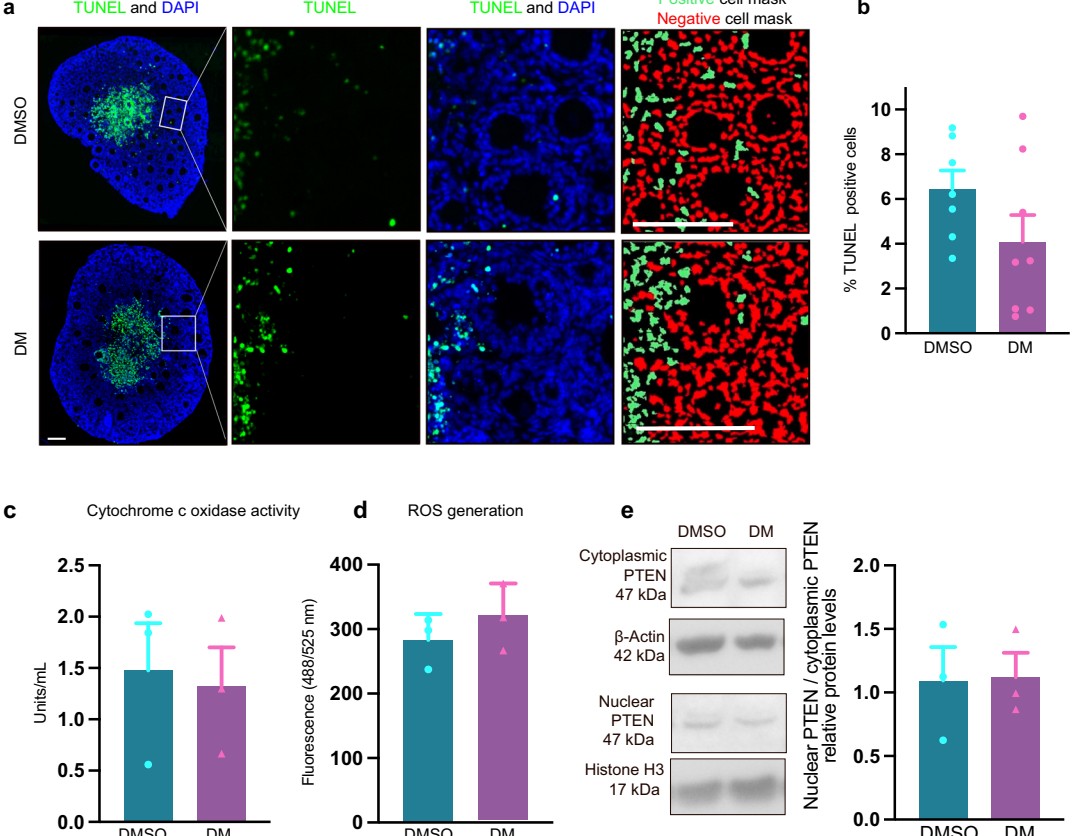

**Fig. 5 | High-quality oocytes after dorsomorphin treatment. a** Apoptotic cells were detected with TUNEL staining and counterstained with DAPI to visualize the ovarian structures. The top images represent a whole ovary and show that most TUNEL stains are to be detected in the medulla. Scale bar: 50 µm. The white squares illustrate representative areas of the ovarian cortex containing follicles. The selected areas are enlarged in the images to the right. Scale bar: 20 µm. Moreover, the images to the right represent how the images were analyzed with automated cell imaging, establishing masks for the number of apoptotic cells (green mask) and viable cells (red mask). **b** Quantification of the percentage of TUNEL-positive cells (apoptotic cells), DMSO $n = 7$ biologically independent samples (ovaries), and DM $n = 8$ biologically independent samples (ovaries). **c** Cytochrome c oxidase activity in ovaries cultured with dorsomorphin or DMSO for seven days, DMSO $n = 3$ biologically independent samples (ovaries) consisting of 4 ovaries/group and DM $n = 3$ biologically independent samples (ovaries) consisting of 4 ovaries/group. **d** Quantification of ROS generation in five-day in vitro cultured ovaries exposed to DMSO or dorsomorphin, DMSO $n = 3$ biologically independent samples (ovaries) consisting of 2 ovaries/group and DM $n = 3$ biologically independent samples (ovaries) consisting of 2 ovaries/group. **e** PTEN protein abundance in the cytoplasmic and nuclear compartment and histone H3 and β-actin were used as loading controls, DMSO $n = 3$ biologically independent samples (ovaries) consisting of 10 ovaries/group and DM $n = 3$ biologically independent samples (ovaries) consisting of 10 ovaries/group. Full-length Western blots are shown in S7. **b–e** The data are the mean ± SEM and were analyzed with an unpaired $t$ test. *$P < 0.05$.

size (DMSO *vs.* DM, day 0: 71.65 ± 2.071 µm *vs.* 70.250 ± 1.763 µm, $p = 0.6708$; day 6: 131.563 ± 10.787 µm vs. 127.468 ± 4.824 µm, $p = 0.7329$; day 12: 266.159 ± 4.391 µm *vs.* 265.080 ± 5.153 µm, $p = 0.8751$) (Fig. 6c). Moreover, assessment of antral follicle development (DMSO: 60.74 ± 1.832% *vs.* DM: 58.93% ± 4.494, $p = 0.7286$) (Fig. 6d) and degeneration of follicles (DMSO: 38.43 ± 23.77% *vs.* DM: 40.08 ± 13.33%, $p = 0.9555$) (Fig. 6e) during long-term culture also showed no significant difference between DMSO− or dorsomorphin-activated follicles. In line with this, we showed no significant difference in the follicle survival rate (DMSO: 76.11 ± 11.29% *vs.* DM: 72.59 ± 6.329%, $p = 0.7994$) (Fig. 6f). Next, COCs were isolated and cultured with hCG and showed that dorsomorphin oocytes were able to resume the first meiotic division, which was examined by extrusion of the first polar body (Fig. 6g). The size of the MII oocytes was, in alignment with this, not affected by dorsomorphin-induced activation (DMSO: 40.87 ± 0.298 µm vs. DM: 40.95 ± 0.2257 µm, $p = 0.8319$) (Fig. 6h), nor was the MII development rate (DMSO: 60.74 ± 1.832% *vs.* DM: 53.37 ± 17.38%, $p = 0.5103$) (Fig. 6i). Taken together, the results underline that dorsomorphin-initiated primordial follicle activation maintained meiotic competence and thus good quality.

## Discussion

Our work illustrates how cellular manipulation of the AMPK pathway can be used to regulate ovarian primordial follicle reserve in vitro. The potential to identify the cellular and molecular mechanisms responsible for AMPK-related primordial follicle regulation has not previously been addressed, and increases our knowledge of the complex network of many factors that interact to govern the important task of correctly regulating the pool of primordial follicles.

After AMPK is phosphorylated, it can then phosphorylate the PGC1a protein, preventing its degradation. In our analysis of human RNA data, we found that the mean FPKM values for the *PPARGCA1* gene were 3.99 in oocytes of primordial follicles and 5.913 in oocytes of primary follicles. Interestingly, based on AMPK transcripts, we might have expected the opposite pattern. However, both transcripts are relatively high, suggesting that PGC1a may play a vital role in the oocytes of both primordial and primary follicles. This underscores the significance of AMPK's function in these early-stage follicles. We noted that the *PRKAA1* and *PRKAA2* genes are expressed in dormant human primordial follicles at high levels, with differential expression as the follicles are activated and become primary.

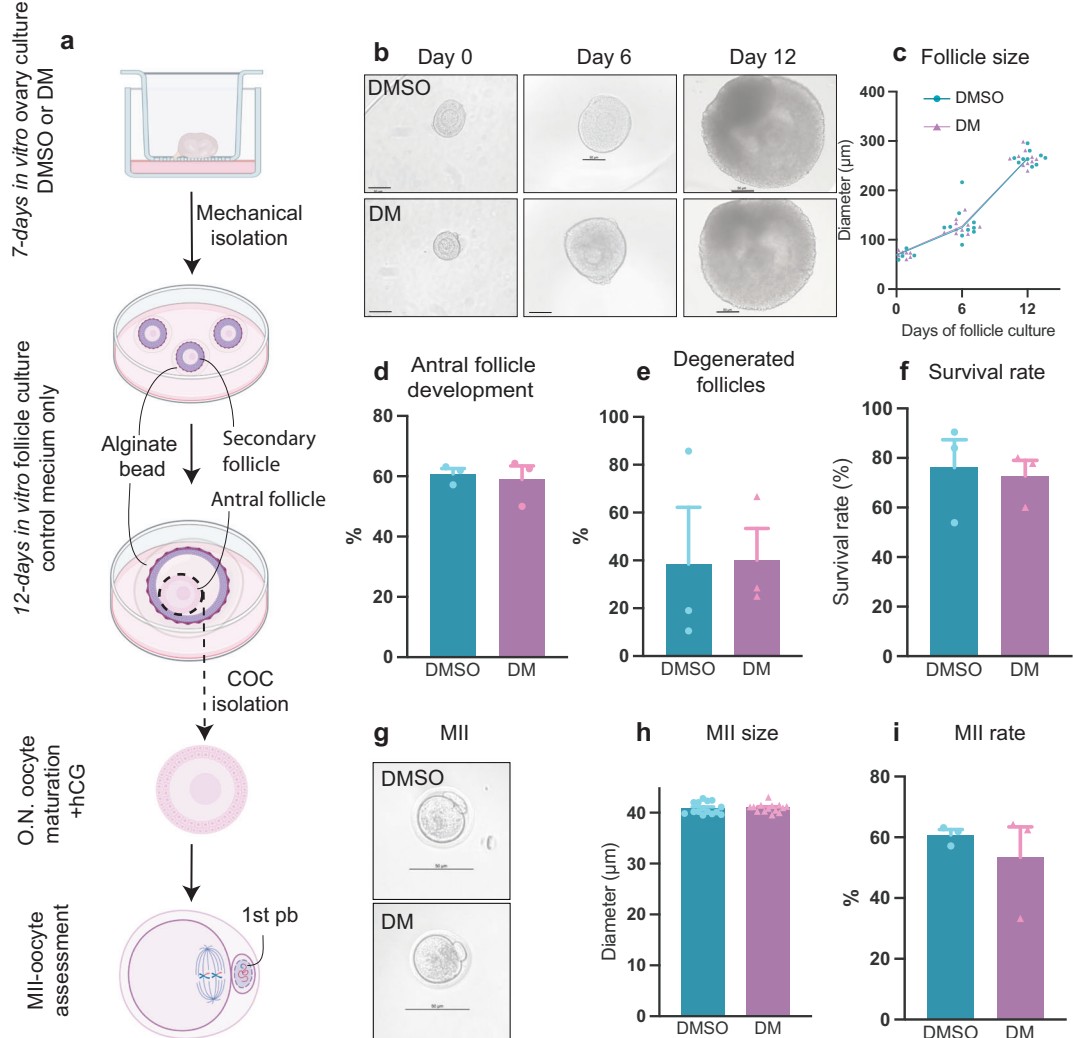

**Fig. 6 | Oocytes from dorsomorphin-treated follicles resume meiosis and form metaphase II oocytes. a** Schematic illustration of the experimental design of the in vitro strategy for the development of mature oocytes from early-stage follicles to MII oocytes. **b** Representative photomicrographs of follicle growth on days 0, 6, and 12 during the 12-day culture in vitro. On day 0, secondary follicles from in vitro cultured ovaries exposed to DMSO or dorsomorphin were isolated and encapsulated in alginate. The follicles were cultured for 12 days in a control medium maintained their three-dimensional structure, and ultimately formed an antrum. Scale bar: 50 μm. **c** Follicle diameters on days 0, 6, and 12 were measured in both groups and increased during the culture period, DMSO $n = 10$ biologically independent samples (ovaries) and DM $n = 10$ biologically independent samples (ovaries). **d** Antrum formation (day 12), DMSO $n = 29$ biologically independent samples (ovaries), and DM $n = 22$ biologically independent samples (ovaries). **e** Degenerated follicles $n = 12$ (day 12). **f** Survival rate DMSO $n = 47$ biologically independent samples (ovaries) and DM $n = 36$ biologically independent samples (ovaries) (day 12). **g** Representative photomicrographs of MII oocytes from DMSO and dorsomorphin-activated follicles, scale bar: 50 μm. **h** MII size, DMSO $n = 15$ biologically independent samples (ovaries) and DM $n = 14$ biologically independent samples (ovaries). **i** MII developmental rate, DMSO $n = 29$ biologically independent samples (ovaries), and DM $n = 21$ biologically independent samples (ovaries). Data are given as the mean ± SEM and were analyzed with an unpaired $t$ test, $*p < 0.05$. **a** created with BioRender.com.

We also noted mRNA expression of genes encoding AMPK downstream effectors, such as PGC1 (encoded by the *PPARGC1A* gene) and ACC protein (encoded by the *ACACA* gene). After AMPK is phosphorylated, it can then phosphorylate the PGC1a protein, preventing its degradation. The *PPARGC1A* transcript is relatively high in early follicles, suggesting that PGC1a may play a vital role in the oocytes of both primordial and primary follicles. When AMPK is activated, it inhibits ACC, reducing malonyl-CoA production and promoting fatty acid oxidation while suppressing fatty acid synthesis. We also analyzed the transcript levels of Malonyl-CoA (MLYDC) downstream and found mean FPKM values of 1.67 in the primordial follicles and 0.24 in the primary follicles. Interestingly, although AMPK activation usually leads to reduced malonyl-CoA production, the MLYCD transcripts remained relatively low in both groups, but the protein levels might have been higher. This emphasizes the pivotal role of AMPK in the function of these early-stage follicles.

To decipher the more precise role of AMPK in the regulation of primordial follicles, AMPK function was manipulated using metformin and dorsomorphin, which activate and inhibit AMPK, respectively. Metformin is an effective ovulation induction agent for nonobese women with polycystic ovary syndrome, where it has been found to reduce the risk of ovarian hyperstimulation syndrome and increase ovulation[44]. We found that metformin reduced the activation of primordial follicles, in line with the effect observed in a chemotherapy-induced in vivo mouse model, where co-treatment of metformin and cyclophosphamide protected the depletion of primordial follicles[26]. This is in contrast to the effect of dorsomorphin, which stimulated primordial follicle activation, in line with a previously published study (Lu et al.), where the authors showed that Compound C (DM) stimulated ovarian follicle growth. As AMPK inhibition using compound C was shown to reduce the phosphorylation of tuberous sclerosis complex 2 (TSC2) in cardiac hypertrophy cells and cancer cells, leading to

phosphorylation of S6K1 in vitro[45], and as activation of mTOR (through S6K1 phosphorylation) is well known to activate primordial follicles, it was hypothesized that inhibition of AMPK could activate primordial follicles through the mTOR pathway[25]. Our study used AMPK inhibition to explore how dorsomorphin affects global gene expression and how this effect is correlated with the safety of oocyte development and quality. As the potential to accelerate the activation of dormant primordial follicles in the ovary has an attractive clinical perspective, the effect of dorsomorphin was submitted to in vitro culture of human tissue, where the effect of dorsomorphin was similar to the effect observed in murine tissue. Collectively, our study establishes that AMPK signaling guards the primordial follicle pool and further suggests that activation of AMPK may be useful to rescue loss of egg reserve, e.g., for patients suffering from POI or patients undergoing chemotherapy in connection with cancer treatment, which would normally deplete the ovarian reserve. In contrast, POI patients or women of advanced maternal age who wish to have a pregnancy could benefit from a novel treatment that awakens their dormant follicles. The patient tissue used in this study was obtained from two patients aged 32 and 49 with different hormone profiles; the tissue donated from the 32-year-old woman had low FSH levels, and the 49-year-old patient had even lower FSH levels, corresponding to expected values in the age range and in alignment with the expected anti-Mullerian hormone (AMH) and sex hormone levels. This strongly indicates that dorsomorphin is broadly effective and not dependent on specific FSH and AMH) levels. However, as metformin has been documented to reduce the circulating level of testosterone in both men and women and in animal models, metformin exposure in utero induced sex-specific reproductive changes, indicating a need for studies exploring the association between metformin exposure and reproductive outcomes in humans[46]. Dorsomorphin is not an approved drug but an experimental compound (as listed in DrugBank), and further development of this compound is needed to evaluate its clinical potential.

Mechanistically, our data suggest that the inhibitory effects of AMPK activation on follicle activation do not involve the PI3K/PTEN/AKT pathway. Instead, our data support a model wherein dorsomorphin inhibits AMPK, where it acts as a gatekeeper for follicles. This resembles the effect of FOXO3a, as well as PTEN, which both have protective roles for the egg reserve, and we suggest that these pathways cooperate to control activation and dormancy of primordial follicle pool. Mechanistically, PTEN reverses phosphorylation of the 3'-OH group on the inositol ring of inositol phospholipids, thus reversing the actions of PI3K[47]. Pharmacological inhibition of PTEN with bpV(OHpic) in human tissue revealed an increase in primordial follicle activation, which led to the conclusion that inhibition of PTEN with 1 µM bpV(HOpic) affects human ovarian follicle development by promoting the initiation of follicle growth and development to the secondary stage but severely compromises the survival of isolated secondary follicles under these conditions[37]. This observation was followed up by the authors in a later study, where inhibition of PTEN was shown to cause DNA damage and reduce DNA repair responses in bovine follicles[38]. In dorsomorphin-treated follicles, we observed no alterations in apoptosis and ROS levels, as well as no changes in PTEN intracellular localization. This finding was confirmed by RNA sequencing of single oocytes laser-dissected from precisely staged primordial and primary follicles, which confirmed that no steroidogenesis genes were induced and that the effect was focused on AMPK signaling genes. The downregulated Stik11 gene, whose product encodes a serine/threonine-protein kinase, acts as a key upstream regulator of AMPK by mediating phosphorylation and activation of the AMPK catalytic subunits PRKAA1 and PRKAA2, confirming the decreased P-AMPK. The upregulation of the Cab39 gene, whose product is Stik11 activation, suggests that the oocyte might compensate for the loss of Strada. The Prkag1 and Prkag2 genes, as well as the Prkab2 gene are upregulated, suggesting that the oocytes compensate for the loss of the alpha and beta subunits. It is curious that we did not detect Prkaa2 in our transcriptome data. We probed for its presence in a study that compared in vivo growth of primordial and primary follicles to that of in vitro growth[48] and found that Prkaa2 was highly upregulated during in vitro growth, which strongly suggests that

dorsomorphin is very effective in inhibiting the Prkaa2 transcript. When evaluating AMPK function by examining its downstream substrates, we observed greater expression of Ppargc1a transcripts in oocytes from primary follicles than in oocytes from primordial follicles. This observation suggests that AMPK predominantly operates as a gatekeeper of the primordial follicle pool, and it is possible that the regulation of the Ppargc1a gene occurs independently of AMPK. Furthermore, there was a slight upregulation in Acaca transcripts in both groups, possibly indicative of a compensatory response aimed at maintaining the energy balance.

WNTs are highly conserved signaling molecules that act through β-catenin-dependent and β-catenin-independent pathways to regulate important processes of cellular growth and differentiation, including cell proliferation[49]. The presence and activity of WNT signaling components in the ovary is not unexpected, given the variety of physiological processes known to be regulated by the WNT family of proteins[50]. However, characterization of specific WNT molecules during folliculogenesis has been focused primarily on Wnt2/WNT2 and Wnt4/WNT4 in mice, rats, and humans[29,50], which is linked to the upregulation of FOXO.

WNTs, highly conserved signaling molecules, have been shown to regulate cellular growth, differentiation, and proliferation, primarily through the Wnt/FOXO pathway. Dorsomorphin-induced AMPK inhibition activates primordial follicles via this pathway, even though increased FOXO3a signaling may take longer to manifest at the protein level. This is in line with our findings that indicate that dorsomorphin-induced AMPK inhibition activates primordial follicles through the Wnt/FOXO pathway. In our immunostaining with FOXO3A, we did not detect increased signaling; however, the upregulation of the FOXO3A transcript was in the lower end, and this might take a longer time to show this effect on protein levels. In summary, our study demonstrates that in vitro-matured oocytes developed from dorsomorphin-treated follicles proceed to meiosis and form metaphase II oocytes, indicating that dorsomorphin did not compromise the potential to generate oocytes. This research opens doors to further understanding the signaling within the ovary that can regulate the ovarian reserve dynamics. Further studies are needed to functionally assess this in relation to the endogenous signaling from the ovary, and the related hypothalamic-pituitary-gonadal axis controlling the reproductive hormones.

## Methods
### Animals
Female C57BL/6JRj (Janvier) mice were mated with CBA/JRj (Janvier, Le Genest-Saint-Isle, France) males to obtain C57BL/6JRj x CBA/JRj F1 hybrid mice in the Biomedical animal facilities at Aarhus University. Mice were housed in a daily 12-hour light/dark cycle in a temperature-controlled environment and provided with food and water ad libitum. Seven- to eight-day-old female F1 mice were used in this study. Animals were handled according to Danish national institutional regulations and approved by the Ethics Committee for the Use of Laboratory Animals at Aarhus University (permit numbers: 2020-15-0201-00757 to KLH).

### Murine in vitro ovarian culture
Mice were sacrificed by cervical dislocation, and ovaries were removed and submitted to ovary in vitro culture[48,51,52]. Then, the ovaries were excised from the surrounding bursa and washed with α-minimal essential medium (α-MEM, Gibco, Scotland, Paisley, UK) supplemented with 10% fetal bovine serum (FBS, Gibco, Scotland, Paisley, UK). The ovaries were placed on culture membranes (0.4 µm pore size, 6.5 mm diameter, Corning, MA, United States) floating on culture medium: α-MEM supplemented with fetal bovine serum (FBS; 10%, Thermo Fisher Scientific, Waltham, Massachusetts, USA), insulin, transferrin and selenium (ITS; 1%, Thermo Fisher Scientific, Waltham, Massachusetts, USA), recombinant follicle-stimulating hormone (FSH; 20 mIU, Sigma, St Louis, MO, USA), penicillin and streptomycin (100 mg/mL and 50 ng/mL, Thermo Fisher Scientific, Waltham, Massachusetts, USA). Additionally, the culture medium was supplemented with DMSO, dorsomorphin (5–10 µM reconstituted in DMSO (lead is 10 µM), Sigma, St Louis, MO, USA), BAY-3827 (10 µM, MedChemExpress,

Monmouth Junction, USA) or metformin (20–100 μM reconstituted in water to 100 mM stock solution and diluted in α-MEM, Sigma, St Louis, MO, USA), and the drug concentration was based on previous studies[25,53–55]. The membranes were inserted into 24-well plates (Sigma, St Louis, MO, USA), and 200 μL culture medium was added beneath the membrane. The ovaries were cultured for 6 h or five to seven days at 37 °C and 5% $CO_2$, and 70 μL of culture medium was changed every other day.

## Human in vitro culture of ovary biopsies

Ovarian cortical tissue was obtained from one patient (32 years old) undergoing oophorectomy prior to gonadotoxic treatment of a malignant disease (unrelated to the ovary) and one patient (49 years old) undergoing prophylactic ovary removal to reduce the risk of developing ovarian cancer. In connection to oophorectomy, a small piece of the ovarian cortex is used for evaluating the ovarian reserve and for research purposes. Written informed consent was obtained from women with low ovarian reserve under a protocol approved by the Ethics Committee of the Faculty of Aarhus University (permit number: 2020-15-0201-00757 to KLH, EHE, EE, MD and OM). Patients consented to the research conducted. Tissue biopsies were washed in Leibovitz's L-15 medium (Gibco, Scotland, Paisley, UK) supplemented with human serum albumin (10% (w/v)) (Merck, Darmstadt, Germany), ITS (1%), penicillin–streptomycin (1%) and recombinant follicle stimulating hormone (300 mIU/mL, Gonal F, Serono, Genéve, Schweiz), and most medullar tissue was removed. Next, the remaining cortex was fragmented to approximately $2 \times 1 \times 1$ mm³, and the fragments were randomly divided into the control or drug-exposed group. The fragments were individually placed in a 96-well V-bottomed culture plate in 250-α-MEM supplemented with human serum albumin (10%), ITS (1%), penicillin–streptomycin (1%) and recombinant follicle stimulating hormone (300 mIU/mL) at 37 °C. Every second day, 100 μL of the culture medium was replaced during a 2-week culture period[52].

## Histological analysis and follicular classification

Ovarian tissue pieces from mice (n = 3–16/group) and humans (n = 2 patients) were harvested at different time points and fixed in paraformaldehyde (PFA; 4%, Merck) overnight at 4 °C. Subsequently, the tissues were dehydrated in ascending concentrations of ethanol (70–99.9%), cleared in xylene, and finally embedded in paraffin wax[48,51,52]. This tissue was cut into five-micron serial sections and stained with haematoxylin and eosin. For murine ovaries, the number of stage-specific follicles was counted and averaged for a minimum of 10 sections per ovary (3rd–5th), and for human tissue pieces, every 5th section was counted. Follicles were classified as primordial, primary, and secondary. A primordial follicle was characterized as an oocyte surrounded by a few flattened granulosa cells. Primary follicles were defined as those with an oocyte encapsulated by one layer of cuboidal granulosa cells, and secondary follicles were defined as oocytes encapsulated by two or more layers of cuboidal granulosa cells. In mouse ovaries only follicles appearing healthy and with a nucleus were counted to avoid recounting the same follicles. In human tissue, patients were only included if it was possible to count 10 or more follicles in each group.

## Western blotting analysis

For total protein extraction, in vitro cultured ovaries were lysed using RIPA buffer containing protease and phosphatase inhibitors (n = 4–6 ovaries/group). For cytoplasmic and nuclear protein extraction, in vitro cultured ovaries were lysed by following the protocol for the CellLytic™ NuCLEAR™ Extraction Kit (NXTRACT, Sigma, St Louis, MO, USA) (n = 10/group). The protein concentration was quantified using BCA quantification and submitted to Western blotting[48,51,52]. For all assays, 20 μg of protein was separated by 4–12% SDS-PAGE and electroblotted onto polyvinylidene difluoride membranes (0.45 μm pore size, Merck Millipore, Burlington, Massachusetts, USA). Membranes were blocked in 5% nonfat milk or 5% BSA reconstituted in Tris-buffered Saline with Tween 20, and incubated with the following primary antibodies at 4 °C overnight: AMPKα (1:1000, 2523, Cell signaling technology, Danvers, Massachusetts, USA), phosphor-

AMPKα (Thr172) (1:500, 2531, Cell signaling technology), Akt (1:1000, 9272, Cell Signaling Technology), Phospho-Akt (Thr308) (1:1000, 9275, Cell Signaling Technology), Phospho-Akt (Ser473) (1:2000, 9271, Cell Signaling Technology), Foxo3a (1:1000, 2497, Cell Signaling Technology), p27 (1:75, sc-1641, Santa Cruz Biotechnology, Dallas, Texas, USA), Bcl-2 (1:500, MAB8272, R&D systems, Minneapolis, Minnesota, USA), Bax (1:500, ab182733, Abcam, Cambridge, UK), PTEN (1:500, ab31392, Abcam), phosphor β-catenin (Ser552) (1:2000, 9566, Cell Signaling Technology) and β-Actin (1:3000, A5441, Sigma, St Louis, MO, USA) served as loading control for total and cytoplasmic protein amount, whereas Histone H3 (1:2000, 4499, Cell Signaling Technology) and anti-TATA binding protein TBP (1:1000, Abcam, ab51841) served as loading control for nuclear protein amount. The following HRP-conjugated secondary antibodies were used: goat anti-rabbit (Thermo Fisher Scientific, 65–6120) and rabbit anti-mouse (Thermo Fisher Scientific, 61–6520). Visualization was conducted using enhanced chemiluminescent substrate for detection of horseradish peroxidase as detection reagent (ECL, ThermoScientific, Massachusetts, USA) and a LAS 4000 imager (He Healthcare, Chicago, Illinois, USA). Finally, the density of each band was evaluated using ImageJ (Version 2.9.0 U.S. National Institutes of Health, Bethesda, Maryland, USA)).

## Immunofluorescence staining

For immunofluorescence assays, sectioned in vitro cultured ovaries (5 μm) (n = 3) were subjected to dehydration and deparaffinization[48,51,52]. Antigen retrieval using heat and citrate buffer (0.01 M) was performed, followed by permeabilization (0.5% Triton 100×) and blocking in donkey serum (10%) (Merck Millipore Chemicon, Burlington, Massachusetts, USA) for one hour at room temperature. Primary antibodies against AMPKα1 (1:50, sc-398861, Santa Cruz Biotechnology), AMPKα2 (1:1000, ab3760, Abcam), p27 (1:50, sc-1641, Santa Cruz Biotechnology), and Foxo3a (1:200, 2497, Cell Signaling Technology) were applied to donkey serum (10%) overnight at 4 °C. Secondary antibody labeling was performed with Alexa flour 555 donkey anti-mouse (1:300, Invitrogen, Waltham, Massachusetts, USA) and Alexa flour 488 donkey anti-rabbit (1:700, Invitrogen) for one hour at room temperature. Sections were counterstained with DAPI (1 μg/μL, Sigma, St Louis, MO, USA). Finally, sections were mounted using Dako fluorescence mounting medium (Agilent Technologies, Santa Clara, CA, USA). Immunofluorescence staining was imaged by an ImageXpress pico automated cell imaging system (PICO image express, Molecular Devices, Sunnyvale, CA, USA), and linear adjustments were conducted using ImageJ (Version 2.9.0). U.S. National Institutes of Health, Bethesda, Maryland, USA).

## TUNEL

The TUNEL assay was performed to detect apoptosis in 5-day in vitro cultured ovaries (n = 7–8) using the in-site Cell Death Detection Kit (Roche, Basel, Schweiz)[48,51,52]. In brief, sections were processed as described for immunofluorescence staining and labeled with TdT for two hours in a humidity chamber at 37 °C. Negative controls were omitted for TdT. The sections were counterstained with DAPI (1 μL/mL, Sigma, St Louis, MO, USA)) and visualized with Dako fluorescent mounting medium. Fluorescent imaging (PICO image express, Molecular Devices, Sunnyvale, CA, USA) was applied to calculate the total number of apoptotic cells versus the total number of cells in representative sections of the cortex.

## Cytochrome c oxidase activation assay

Cytochrome c oxidase activation was measured using a spectrophotometer (n = 4 ovaries/group). First, the mitochondrial fraction was isolated using a mitochondria isolation kit (Sigma, St Louis, MO, USA)[48]. The activity of cytochrome c oxidase was measured by a cytochrome c oxidase assay kit (CYTOCOX1, Sigma, St Louis, MO, USA) and applied as described by the manufacturer. In brief, it is a colorimetric assay based on changes in the absorbance of cytochrome c at 550 nm depending on its oxidation state. First, a cytochrome c solution was prepared (0.22 mM) and reduced with DTT (0.1 mM). By adding the isolated ovarian mitochondrial fraction, cytochrome c was again reoxidized by cytochrome c oxidase from the

mitochondrial fraction and measured by a spectrophotometer (Ultrospec 2000, Pharmacia Biotech, Uppsala, Sweden) using an extinction coefficient $OD_{550}$. To convert absorbance into specific activity (expressed in units $mL^{-1}$), we used the equation $\Delta A/\Delta t \times V \times$ dilution factor of enzyme/$(P_{total} \times \varepsilon)$, where $\Delta A$ is the increase in absorbance at 550 (absolute value), $\Delta t$ is the reaction time (minutes), $V$ is the reaction volume (mL) in the cuvette, $P_{total}$ is the total volume of mitochondrial protein in the cuvette (mL) and $\varepsilon$ is the extinction coefficient between ferrocytochrome c and ferricytochrome c at 550 nm ($21.84 \, mmol^{-1} \, cm^{-1}$).

### Detection of ROS production
Spectrofluorometry was used to measure ROS production[48]. After ovaries were cultured in vitro with DMSO or dorsomorphin for five days ($n = 2$ ovaries/group in three repeats), ovaries were washed in PBS. Ovaries were then incubated with 2'7'-dihydrodichlorofluorescein diacetate (DCFH-DA) (1 µL/mL, Sigma, St Louis, MO, USA) for 45 min at 37 °C before the ovaries were again washed in PBS. Subsequently, the ovaries were lysed (Tris-HCl (10 mM), EDTA (20 mM) and Triton x-100 (0.25%)) and centrifuged at 4 °C at $10,000 \times g$ for 20 min. The supernatants were loaded into a black 96-well plate, and fluorescence was monitored using spectrofluorometry (488 nm excitation and 525 nm emission) (CLARIOstar, BMG labtech, Ortenberg, Germany).

### Isolation, encapsulation and in vitro culture of secondary follicles
On day seven of in vitro ovarian culture, secondary follicles were mechanically isolated using fine needles (n > 48/each group) under a stereomicroscope. The secondary follicles were encapsulated using sodium alginate[48,51]. Isolated secondary follicles were individually placed in droplets of sodium alginate (1% (w/v)) mixed with α-MEM and encapsulated by slowly pipetting the follicle and alginate solution into a cross-linking solution (50 mM $CaCl_2$ and 140 mM NaCl). Encapsulated follicles were placed in drops of culture medium consisting of α-MEM supplemented with FBS (10%), ITS (1%), FSH (100 mIU/mL, GONADAL-f), penicillin and streptomycin (100 mg/mL and 50 ng/mL) in embryo dishes and overlaid with mineral oil. Follicles were cultured for 12 days at 37 °C and 5% $CO_2$, and every other day, half of the culture medium was replenished to maintain the culture environment. After plating, encapsulated follicles were imaged using Leica Microsystems (BMI4000B, Wetzlar, Germany) and again on days 6 and 12. The follicular size was obtained by averaging two perpendicular measurements of the follicle diameter (n = 14-15) using ImageJ (Version 2.9.0). U.S. National Institutes of Health, Bethesda, Maryland, USA).

### Oocyte maturation
On the 12th day of follicle in vitro culture, antral follicles were mechanically released from sodium alginate beads[48,51]. Next, from the antral follicles, cumulus cell-enclosed complexes (COCs) were mechanically isolated without damaging the oocyte. The isolated COCs were incubated in drops with α-MEM supplemented with FBS (10%) ITS (1%), follitropin alfa (100 mIU/mL, GONAL-f) penicillin (100 mg/mL), streptomycin (50 ng/mL) and hCG (10 IU/mL) (Sereno) overnight to induce ovulation. The following day, oocytes with metaphase II development competences were individually evaluated according to the presence of first polar body extrusion.

### Laser capture microdissection
Ovaries cultured in vitro for five days with DMSO or DM were fixed using 4% PFA at 4 °C, dehydrated, paraffin embedded, and stored at −80 °C[48]. Next, the sections were sectioned and mounted on membrane glass slides (Arcturus™ PEN Membrane Glass Slides, Applied Biosystems, Life Technologies, Foster City, CA, United States) and stained with H&E. On the same day, oocytes (n = 200/group) of primordial and primary follicles in both groups were isolated by LCM (LMD7 laser, Leica Microsystems, Wetzlar, Germany) based on the morphological appearance[48]. The laser

beam steers along the desired cut line around the oocyte in the tissue, and the isolated oocyte was collected into the lid of a PCR tube by gravity.

### RNA extraction
The LCM-collected oocytes were centrifuged from the lid into the collection tube, and Arcturus™ paradise™ extraction was used[48]. An isolation kit (#KIT03121, Arcturus Bioscience Inc., Mountain View, CA, United States) was used to isolate total cellular RNA[48]. RNA from each cap was pooled for the same oocyte stage during extraction. Total RNA from the oocytes was isolated using Picopure (#KIT0204 Arcturus Bioscience, Inc. CA, United States).

### Library preparation and sequencing
The ovation PICO DL WTA system V2 RNA amplification system (#3312 NuGen, Inc., San Carlos, CA, United States) was utilized to convert isolated RNA to cDNA and linear amplification. Next, library preparation was conducted using an Illumina TruSeq RNA Access kit, according to www.BGI.com. RNA sequencing was performed on an Illumina HiSeq platform with two lanes (5 Gb per sample). Library preparation for collected oocytes was performed using the Smart-seq II method, and RNA sequencing was performed using the Illumina HiSeq platform with one lane (5 Gb per sample) (www.BGI.com)[48].

### Transcriptome analysis
The raw data were quality filtered and trimmed, and adaptor sequences were removed using trim_galore (v0.4.1) (performed by https://omiics.com/). FastQC was applied for quality control. Differential expression analysis was carried out using mapping the filtered reads to the mouse genome (mm10) using TopHat2 (performed by https://omiics.com/). FeatureCounts software was then applied to quantify the number of reads mapped to each gene using gene annotation from Ensembl release 19. Then, DESeq2 in R was applied for differential expression analysis, and normalization was applied by DESeq2 to permit direct comparison between the samples (https://omiics.com/). A heatmap was created using Prism (version 9) to visualize up- and downregulated SDEGs.

### Statistics and reproducibility
Graphs and statistical analyses were completed with GraphPad PRISM 9 (GraphPad Software Inc., San Diego, CA, USA). The statistical results of all experiments represent the mean of at least three independent replicates, and error bars are standard errors of the mean (SEM). The exact number of experimental results are pooled from are listed in the figure legend and visualized on all graphs. Statistical significance was compared between a pair of datasets with at least three independent biological replicates by unpaired two-tailed $t$ test. A $P$ value < 0.05 was considered significant. For experiments where $n > 3$, the assumption of normality was checked by a QQ plot, and to test that the observations came from the same distribution, an $F$ test was conducted. One-way ANOVA was performed on data containing three or more data sets, followed by a Bonferroni means comparison. Normality was validated by QQ plots, and the assumption of equal homogeneity was checked with Barlett's test of equal variance.

### Reporting summary
Further information on research design is available in the Nature Portfolio Reporting Summary linked to this article.

### Data availability
All RNA-seq data are available at GEO (accession number: GSE230258). Link: https://www.ncbi.nlm.nih.gov/geo/query/acc.cgi?acc=GSE230258. Source Data for Figs. 1a, 3b, and 4a are available in Supplementary Table 1. Uncropped western blottings are available in Supplementary Figs 4, 6, and 7.

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

## Acknowledgements

We thank Anders Heuck for excellent technical help and members of the KLH lab. We thank Dr. Med Professor Ole Mogensen, Department of Obstetrics and Gynecology, Aarhus University Hospital, DK-8000 Aarhus C, Denmark, for good medical discussions. The work is supported by grants from the Novo Nordic Foundation (NNF16 °C0022480 and NNF17 °C0026820 to K.L.H.), the Graduate School of Health, Aarhus University (to K.L.H.), the Augustinus Foundation (17-4844 to KLH), the Aase and Ejnar Danielsen fond (18-10-0470 to K.L.H.), the Aarhus University Research Fond (AUFF-E-2020-9-11), and the Carlsberg fond (CF18-0474 to M.A.).

## Author contributions

J.F.M. and K.L.H. designed the study, designed and performed experiments, and analyzed and interpreted the data. J.F.M. performed all cellular and biochemical assays, mouse and human follicle in vitro assays, L.C.M. and RNA collection, immunohistochemistry, western blotting, and in vitro and ex vivo culture of murine and human tissue. J.F.M. and M.A. performed in vitro maturation of follicles to mature metaphase eggs. J.F.M. and K.L.H. analyzed RNA data. E.H.E., E.E., and M.D. enrolled human participants and provided the ovarian biopsies from patients. J.F.M. drafted all the figures. J.F.M. and K.L.H. wrote the manuscript, and all authors approved the final version.

## Competing interests

E.H.E. and K.L.H. are inventors on a patent relating to the activation of primordial follicles in mammals filed by Aarhus University, D.K. All other authors declare no competing interests.
