## [Peer Review File · Communications Biology]

Reviewers' comments:

Reviewer #1 (Remarks to the Author):

Reviewer's response

The manuscript 'Dorsomorphin inhibits AMPK action, upregulates the Wnt and Foxo genes and promotes dormant follicle activation' has been thoroughly read. The study highlights the role of dorsomorphin and metformin in Folliculogenesis via AMPK pathway. The study holds the great importance in the field of infertility and needs to address the following comments prior publication-

1. The manuscript needs to be checked by some English expert for grammatical and expression errors.
2. In the introduction section, the concluding lines need to be reframed with more focus on objectives and future aspects of the study instead of discussing the results.
3. In methodology section-
 - i. Mention the basis and cite references on which the doses of dorsomorphin and metformin was selected.
 - ii. Mention the sample size for each group (animal number, tissues sample no. etc.) taken for the test.
 - iii. Write the names of the city and country of the institutes mentioned or the manufacturers of instruments/kits
 - iv. For each of the protocol mention suitable references.
 - v. Briefly discuss the parameters of oocyte maturation and RNA extraction methodology under their respective subheadings.
4. The result section is well presented. Figure 2c, write in the images the type of the classified follicle.
5. Discussion is also well written but mention how oxidative stress and apoptosis affects folliculogenesis in detail. Conclusion needs to be effectively written emphasizing more on the outcome, significance and future aspects of the study.

Refer the literature:

- Bhardwaj, J.K., and Saraf, P. (2021). Ameliorating potentials of N-acetyl-L-cysteine against methoxychlor instigated modulation in structural characteristics of granulosa cells of caprine antral follicles. *Indian Journal of Biochemistry and Biophysics*, 58, 365-371
 - Bhardwaj, J.K. and Saraf, P. (2015). Granulosa cell apoptosis by impairing antioxidant defense system and cellular integrity in caprine antral follicles post malathion exposure. *Environ Toxicol.* 31(12), 1944-1954.
 - Bhardwaj, J.K., Palliwal, A., Saraf, P., Sachdeva, S.N., (2022). Role of autophagy in follicular development and maintenance of primordial follicular pool in the ovary. *Journal of Cellular Physiology*, 237, 1157-1170
 - Bhardwaj, J.K. and Sharma, R.K. (2011). Scanning electron microscopic changes in granulosa cells during follicular atresia in caprine ovary. *J. Scanning.* 33, 21-24
6. Follow the common pattern of writing references as per the general guidelines. The name of the journal should be either in full or abbreviated, page range is missing in a number of journals and write doi number in all or none.

Reviewer #2 (Remarks to the Author):

In this paper, the authors reported that metformin or dorsomorphin can inhibit or promote activation of primordial follicles in cultured mouse ovaries and human ovarian cortical tissues. The authors claimed that their results identify the AMPK pathways as a potential therapeutic target for infertile patients.

At the same time that this paper is of an interesting topic, the paper was carelessly written and the quality of the paper suffer greatly from unprofessionally presented experimental results. The authors seemed don't know how to prepare a research paper on ovaries. The main conclusion is not supported by the results and almost all morphological results suffer from low quality and a complete ignorance of good morphological photos for a scientific paper. I had never seen any paper using pictures showing only one ovarian follicle and then claim that they saw any changes presented with bar chart. Show us the more or less activated follicles in sections of the whole mouse ovaries or the whole human ovarian tissue, please.

A few specific points are listed below.

1. According to Table 1, metformin inhibited the activation of primordial follicles. However, in Fig.2b and 2d, there were no significant differences observed after metformin and dorsomorphin treatment.
2. When reporting data in mouse ovaries, the authors should show pictures of the entire ovary for example in Fig. 2b and 2d, instead of a partial one.
3. In Fig. 2S, there was no obvious difference observed between the metformin treatment at different concentrations and the control group, which is inconsistent with the author's description.
4. The authors claimed that the ovarian phenotype picture was in Fig. 1, but it should be Fig. 2 (Line 112-116).
5. In Fig. 2g, the structures shown are not typical forms of human follicles. It is recommended that the authors use MVH and Foxl2 staining to observe and confirm if these structures are indeed human follicles.
6. The citation of many references is incorrect, such as reference 10, 12, 24, 25 and 26.
7. Previous studies have indicated that in vitro culture with dorsomorphin in murine ovaries followed by in vivo grafting resulted in more preantral follicles. The authors should discuss the already published paper of the same topic with solid data.

Reviewer #3 (Remarks to the Author):

Overview

In this manuscript, the author showed that Dorsomorphin treatment, a known direct AMPK inhibitor, results in primordial follicle activation in in-vitro systems. The author showed that Dorsomorphin asserts its follicle activation effects directly through AMPK inhibition, not the PI3K-AKT axis. Finally, oocyte developed through Dorsomorphin treated follicle is equivalent to untreated condition.

The manuscript is well-written, and the data is well-represented. Moreover, I thank the author for including full-length Western blots in the Supplementary figures, which makes the review process easier. On the findings, the effect of Dorsomorphin on primordial follicular activation is strong. However, it is not novel since Lu et al. (Lu et al., 2017) have shown the effect of Dorsomorphin on follicular maturation, both in-vivo and in-vitro settings. The novel aspect of this manuscript is the functional study of oocytes derived from Dorsomorphin-treated ovaries, which is shown to retain meiotic competence.

Apart from the lack of novelty, I have a few major and minor comments below.

Major

1. I am convinced that AMPK is expressed in both the primordial and primary follicle and that AMPK expression is upregulated in the primordial follicle compared to the primary follicle. However, AMPK expression itself is not a good estimate of AMPK function. I suggest the author include the expression of AMPK downstream effectors such as PGC1a, GLUT1-4, ACC, and PEPCK as surrogates of AMPK function, especially in Figures 1A and 3B.
2. Does Dorsomorphin change the number of total follicles at all? Or it just changes the distribution from primordial follicles to more mature follicles. Could the author give some estimating parameters of total follicles in the ovarian cortex?
3. Why the western blot in this manuscript are pretty faint? Is it because it represents only a few ovaries per lane?
4. The AMPK role in follicle maturation is not clear. Specifically, in Figure 3, the author claims that Dorsomorphin treatment results in AMPK inhibition which results in more follicular maturation. And in Figure 1, the author also showed that primary follicles have lower AMPK expression, and possible lower AMPK activity. It is possible that Dorsomorphin induces follicles through an AMPK-independent mechanism, resulting in more maturation of follicles and, thus, less AMPK activity in the whole ovary. To delineate this problem, I suggest the author use other AMPK direct inhibitors, such as BAY-3827, and perform the same experiment as in Figure 2 and Figure 3. If the finding is still the same as Dorsomorphin, then it is more likely that AMPK mediates the follicle maturation effect of Dorsomorphin.
5. I am confused with Western blot in Figure 4E and 4F, and related supplementary figures. Are these Western blot laser-captured follicles or the whole ovary? The reason that I am confused is that the author showed that Dorsomorphin increases follicle maturation, which means that if the whole ovary is used for Western blot, then we should see more FOXO3a and p27 in the cytosolic compartment from a higher fraction of matured follicles. If I understand the author's point correctly, the message of this figure is to show that Dorsomorphin activates primordial follicles in a

PI3K/AKT-independent manner. In this case, the author might want to show that the FOXO3a cytosolic to nuclear ratio in the primordial and primary follicle is not statistically different between DMSO control and Dorsomorphin treated, instead of fractionated Western blot.

Minor

1. Figure 1: The label is wrong; b should be moved to the heat map itself. The label 'd' is missing from the figure. Alternatively, include the heatmap description in Figure 1a description.
2. Line 112: Change figure reference from Figure 1b to Figure 2b.
3. Line 596: What is S2 reference to here? Does the author mean Figure S2?
4. Line 113 – 114: Change figure reference from Figure 1C to Figure 2C
5. Line 115: Change figure reference from Figure 1D to Figure 2D
6. Line 116: Change figure reference from Figure 1E to Figure 2E
7. Figure 4C: What is 'Tr', is the author pointing to granulosa cells? Can the author add this to the Figure description?
8. Line 675: in Figure S1 description, change FPRM to FPKM.

Reference

1. Lu, X., Guo, S., Cheng, Y., Kim, J., Feng, Y., & Feng, Y. (2017). Stimulation of ovarian follicle growth after AMPK inhibition. *Reproduction*, 153(5), 683-694. Retrieved Aug 10, 2023, from <https://doi.org/10.1530/REP-16-0577>

Reply to reviewers

Reviewers' comments:

Reviewer #1 (Remarks to the Author):

Reviewer's response

The manuscript 'Dorsomorphin inhibits AMPK action, upregulates the Wnt and Foxo genes and promotes dormant follicle activation' has been thoroughly read. The study highlights the role of dorsomorphin and metformin in Folliculogenesis via AMPK pathway. The study holds the great importance in the field of infertility and needs to address the following comments prior publication.

We are grateful to the reviewer for the appreciation of the study and for the comments and suggestions. Please see below for specific replies.

1. The manuscript needs to be checked by some English expert for grammatical and expression errors.

Before submission, the manuscript was submitted to the editing service offered by Nature.

It was a surprise to have this comment, since we paid attention to the writing (not being ourselves native English speakers). We contacted the Nature editing service and informed them of this comment and after revision, we again submitted it for editing from the Nature service. We hope very much that it stands out in its expert English correction of grammatical and expression errors after this additional round of Nature text editing service.

2. In the introduction section, the concluding lines need to be reframed with more focus on objectives and future aspects of the study instead of discussing the results.

That is a good suggestion, thank you for suggesting this. We have rephrased the conclusion lines of the introduction section and believe they highlight the objectives and future aspects.

3. In methodology section-

i. Mention the basis and cite references on which the doses of dorsomorphin and metformin was selected.

We thank the reviewer for the comment. We conducted initial titration assays (n=3) for both drugs to determine the optimal concentration, and we have referenced relevant sources in the manuscripts. For dorsomorphin, we have included citations to Carson and Kallen (1) and Lu, Guo (2), while for metformin, we have referenced Ma, Wei (3) and Will, Palaniappan (4).

ii. Mention the sample size for each group (animal number, tissues sample no. etc.) taken for the test.

Thank you for pointing this out. Sample sizes have been added to the Methods section, and they are also stated in the figure legends to provide this information as appropriate.

iii. Write the names of the city and country of the institutes mentioned or the manufacturers of instruments/kits

Thank you; this has been added to the manuscript.

iv. For each of the protocols, mention suitable references.

We agree, and we have added suitable references for each protocol in the Methods section.

v. Briefly discuss the parameters of oocyte maturation and RNA extraction methodology under their respective subheadings.

We have added to this section in the Methods to elaborate on the parameters and procedure.

4. The result section is well presented. Figure 2c, write in the images the type of the classified follicle.

In Figure 2, we have categorized the various stages of follicles in the representative ovarian section for both control groups, as seen in the higher magnification.

5. Discussion is also well written but mention how oxidative stress and apoptosis affects folliculogenesis in detail. Conclusion needs to be effectively written emphasizing more on the outcome, significance and future aspects of the study.

We agree and have rephrased the conclusion part of the discussion (at the end).

Refer the literature:

- Bhardwaj, J.K., and Saraf, P. (2021). Ameliorating potentials of N-acetyl-l-cysteine against methoxychlor instigated modulation in structural characteristics of granulosa cells of caprine antral follicles. *Indian Journal of Biochemistry and Biophysics*, 58, 365-371
- Bhardwaj, J.K. and Saraf, P. (2015). Granulosa cell apoptosis by impairing antioxidant defense system and cellular integrity in caprine antral follicles post malathion exposure. *Environ Toxicol.* 31(12), 1944-1954.
- Bhardwaj, J.K., Palliwal, A., Saraf, P., Sachdeva, S.N., (2022). Role of autophagy in follicular development and maintenance of primordial follicular pool in the ovary. *Journal of Cellular Physiology*, 237, 1157-1170
- Bhardwaj, J.K. and Sharma, R.K. (2011). Scanning electron microscopic changes in granulosa cells during follicular atresia in caprine ovary. *J. Scanning.* 33, 21-24

6. Follow the common pattern of writing references as per the general guidelines. The name of the journal should be either in full or abbreviated, page range is missing in a number of journals and write doi number in all or none.

Thank you for spotting this. We have ensured correct formatting in the revision.

Reviewer #2 (Remarks to the Author):

In this paper, the authors reported that metformin or dorsomorphin can inhibit or promote activation of primordial follicles in cultured mouse ovaries and human ovarian cortical tissues. The authors claimed that their results identify the AMPK pathways as a potential therapeutic target for infertile patients.

At the same time that this paper is of an interesting topic, the paper was carelessly written and the quality of the paper suffer greatly from unprofessionally presented experimental results. The authors seemed don't know how to prepare a research paper on ovaries. The main conclusion is not supported by the results and almost all morphological results suffer from low quality and a complete ignorance of good morphological photos for a scientific paper. I had never seen any paper using pictures showing only one ovarian follicle and then claim that they saw any changes presented with bar chart. Show us the more or less activated follicles in sections of the whole mouse ovaries or the whole human ovarian tissue, please.

We are grateful to the reviewer for the comments and suggestions. We were surprised that the reviewer found the writing careless, as several experts read the manuscript, and moreover, we had the Nature editing service work on the manuscript prior to the submission process.

We paid attention to the writing (not being ourselves native English speakers). We contacted the Nature editing service and informed them of this comment and after revision, we again submitted it for editing from the Nature service. We hope very much that it stands out in its expert English correction of grammatical and expression errors after this additional round of Nature text editing service.

We believe it has been substantially improved after the revision.

We did extensive work to add to the morphological part of the manuscript. Although we did not have complete ignorance of the morphology, we were extremely careful to monitor authentic images and use state-of-the-art microscopic equipment.

Please see below for specific replies. We have accommodated all comments and substantially improved the manuscript.

A few specific points are listed below.

1. According to Table 1, metformin inhibited the activation of primordial follicles. However, in Fig.2b and 2d, there were no significant differences observed after metformin and dorsomorphin treatment.

From a partial ovary view, we cannot distinguish between the number of activated and dormant follicles; only the follicle morphology is visible. We selected images with primordial, primary, and secondary follicles to assess their health based on morphology. Therefore, these images validate follicular morphology posttreatment. Consequently, we acknowledge the need to emphasize that these images serve to validate follicular morphology following the specified treatment.

As a supplement, images of the entire ovary have been added to Figure 2d:

The whole ovary images show fewer growing follicles in the metformin-exposed ovary cortex, as it is pinpointed that the cortex only contains primordial follicles. In contrast, the dorsomorphin-exposed ovaries display almost all activated oocytes across the cortex and medulla, revealing a clear difference.

However, discerning a significant difference from a single image can be a challenging task. To ensure a robust assessment, we conducted a comprehensive count of the whole ovaries. Alternatively, we could have opted for weighing the ovaries posttreatment; however, previous research from our laboratory has indicated substantial variations in results using this method. As a result, the most accurate and reliable approach for evaluating the distribution of primordial-to-primary follicles is to meticulously count the number of follicles across the entirety of the ovaries, which is the approach we employed.

2. When reporting data in mouse ovaries, the authors should show pictures of the entire ovary for example in Fig. 2b and 2d, instead of a partial one.

Thank you for this comment. We agree. We have now included a full ovary picture in Figure 2. The image with only a few follicles was meant for readers to assess different follicular stages, especially identifying healthy-looking follicles with intact oocytes and well-organized granulosa cells without pyknotic bodies. We selected images from the broadest part of the ovary to ensure representation, making it challenging to distinguish activated from dormant follicles in a single image.

3. In Fig. 2S, there was no obvious difference observed between the metformin treatment at different concentrations and the control group, which is inconsistent with the author's description.

Thank you for pointing this out. We have inserted new images. However, we would like to emphasize that it can be very difficult to determine the follicular distribution from just one image.

4. The authors claimed that the ovarian phenotype picture was in Fig. 1, but it should be Fig. 2 (Line 112-116).

Thank you for spotting this, the mistake will be corrected in the manuscript.

5. In Fig. 2g, the structures shown are not typical forms of human follicles. It is recommended that the authors use MVH and Foxl2 staining to observe and confirm if these structures are indeed human follicles.

Thank you for the suggestion. Culturing human ovaries in vitro involves using large pieces of ovarian cortex to generate a high number of sections that need to be counted and stained. Notably, most of these sections do not contain follicles; they consist of cortical tissue. Since we cannot determine whether a section contains follicles until it is stained with H&E, conducting immunofluorescence on all the slides would have been impractical. Hence, we opted to use H&E staining. Since we stained our slides with H&E, it was impossible to perform immunofluorescence staining on these slides. However, we have carefully examined substantially more slides and recorded additional follicles that we have now included for illustration in the revised version.

Additionally, although ovarian culture techniques have advanced over recent years, there are still challenges associated with this technique, including maintaining tissue viability and function during and after culture(5, 6).

6. The citation of many references is incorrect, such as reference 10, 12, 24, 25 and 26.

We have thoroughly reviewed the references and are confident that all of them have been accurately cited within the manuscript.

7. Previous studies have indicated that in vitro culture with dorsomorphin in murine ovaries followed by in vivo grafting resulted in more preantral follicles. The authors should discuss the already published paper of the same topic with solid data.

Thank you for suggesting further expanding this part.

True, dorsomorphin was previously shown to stimulate follicle growth (Lu et al 2016) (which focuses on the mTOR pathway, using different approaches than this study), and part of our study independently confirmed this effect to activate primordial follicles. We have expanded our discussion of this previously published paper in the discussion section to emphasize our focus and further on oocyte quality and the effect of dorsomorphin on human tissue, none of which has previously been addressed.

Reviewer #3 (Remarks to the Author):

Overview

In this manuscript, the author showed that Dorsomorphin treatment, a known direct AMPK inhibitor, results in primordial follicle activation in in-vitro systems. The author showed that Dorsomorphin asserts its follicle activation effects directly through AMPK inhibition, not the PI3K-AKT axis. Finally, oocyte developed through Dorsomorphin treated follicle is equivalent to untreated condition.

The manuscript is well-written, and the data is well-represented. Moreover, I thank the author for including full-length Western blots in the Supplementary figures, which makes the review process easier. On the findings, the effect of Dorsomorphin on primordial follicular activation is strong. However, it is not novel since Lu et al. (Lu et al., 2017) have shown the effect of Dorsomorphin on follicular maturation, both in-vivo and in-vitro settings. The novel aspect of this manuscript is the functional study of oocytes derived from Dorsomorphin-treated ovaries, which is shown to retain meiotic competence.

Apart from the lack of novelty, I have a few major and minor comments below.

We would like to express our sincere gratitude to the reviewer for their appreciation of our study and for providing valuable comments and suggestions.

Dorsomorphin's potential to stimulate follicle growth was previously demonstrated in the work of Lu et al. in 2016, with their primary focus on the mTOR pathway, employing different methodologies from those in our study. Our research independently validated the effect of dorsomorphin on the activation of primordial follicles. Furthermore, we enhanced our investigation by introducing several novel aspects that had not been previously described. These additions encompassed the examination of dorsomorphin's impact on human tissue, RNA-seq data analysis and the establishment of a mechanistic link to the Wnt and β -catenin pathways. These novel findings significantly contribute to the understanding of the role of AMPK in follicle development and maturation.

In our revised discussion section, we have placed a more significant emphasis on our focus on oocyte quality and the effects of dorsomorphin on human tissue, which had not been previously addressed in the literature.

We further addressed all comments made by the reviewer; please see below for specific replies.

Major

1. I am convinced that AMPK is expressed in both the primordial and primary follicle and that AMPK expression is upregulated in the primordial follicle compared to the primary follicle. However, AMPK expression itself is not a good estimate of AMPK function. I suggest the author include the expression of AMPK downstream effectors such as PGC1a, GLUT1-4, ACC, and PEPCK as surrogates of AMPK function, especially in Figures 1A and 3B.

Thank you; this is a very good suggestion. We looked at several AMPK effects from our RNA sequencing after dorsomorphin treatment.

We interrogated the human RNA-seq data (Ernst et al HR, 2017) for the presence of genes encoding PGC1a, GLUT1-4, ACC, PEPCK, and downstream substrates for figure 1 (please see table below).

Gene name	Primordial follicle			Primary follicle		
	Means	t.values	p values	Means	t.values	p values
PPARGC1A	4,00	1,99	0,18	5,91	40,36	0,00
PPARG	1,21	1,12	0,38	-	-	-
NRF1	0,60	1,43	0,29	-	-	-
ACACA	4,72	4,07	0,06	3,91	3,67	0,07
MLYCD	1,68	1,76	0,22	0,25	1	0,42
SLC2A1	0,13	1,97	0,19	-	-	-
SLC2A2	-	-	-	1,94	1	0,42
SLC2A3	0,24	2,81	0,11	0,06	1	0,42
SLC2A4	0,03	1	0,42	-	-	-
PEPCK	-	-	-	-	-	-

- PGC1a

In our analysis of human RNA data, we found that the mean FPKM values for the *PPARGCA1* gene were 3.99 in oocytes of primordial follicles and 5.913 in oocytes of primary follicles. Interestingly, based on AMPK transcripts, we might have expected the opposite pattern. However, both transcripts are relatively high, suggesting that PGC1a may play a vital role in the oocytes of both primordial and primary follicles. This underscores the significance of AMPK's function in these early-stage follicles.

To delve deeper, when PGC1a is activated, it acts as a coactivator for transcription factors such as PPARs (peroxisome proliferator-activated receptors) and NRF-1 (Nuclear Respiratory Factor 1). Transcripts of PPARG (1.21) and NRF1 (0.59) were detected only in the oocytes of primordial follicles; however, both had relatively low FPKM levels. This might suggest an increased AMPK function in the oocytes of primordial follicles compared to those in primary follicles. However, note that this refers to the mRNA levels and might not reflect protein levels.

- ACC

We examined transcripts of the ACACA gene (coding for ACC protein) in oocytes from primordial and primary follicles, revealing FPKM values of 4.717 and 3.909, respectively. These relatively high values indicate the role of AMPK in early-stage follicles.

We also analysed the transcript levels of Malonyl-CoA (MLYDC) downstream and found mean FPKM values of 1.67 in primordial follicles and 0.24 in primary follicles. Interestingly, although AMPK activation usually leads to reduced malonyl-CoA production, MLYCD transcripts remained relatively low in both groups. These results may suggest that this pathway is not critical in primordial follicle activation. However, in more advanced follicular stages, we know that beta-oxidation becomes crucial for energy production within follicles.

- Expression of *PEPCK* was not found in the human transcriptome analysis.

- It appears that *SLC2A1-4* genes do not have a great influence in the oocytes of primordial and primary follicles, as the expression levels are very low or not even detected.

From these data, we incorporated the relevant AMPK downstream effectors *PPARGC1A* and *ACACA* into Figure 1 and elaborated on this in the discussion.

For **figure 3B**, we interrogated the RNA data of laser-captured oocytes exposed to DM or DMSO (please see table below).

Gene name	Oocyte of Primordial follicle			Oocyte of Primary follicle		
	BaseMean	Log2foldchange	p.values	BaseMean	Log2foldchange	p.values
Ppargc1a	22,71	2,86	NA	22,71	16,57	NA
Pparg	0,32	-3,20	0,46	0,32	0	1
Nrf1	1922,06	2,64	NA	1922,06	3,72	NA
Acaca	14309,15	4,10	0,02	14309,15	4,81	0,01
Mlycd	10,62	0,25	0,93	10,62	-4,78	0,10

Figure 3B

In primordial follicles, *Ppargc1a* and *Acaca* are expressed, and as we now included them in Figure 1 heatmap, we likewise included them in the heatmap in Figure 3 and elaborated on this in the discussion.

2. Does Dorsomorphin change the number of total follicles at all? Or it just changes the distribution from primordial follicles to more mature follicles. Could the author give some estimating parameters of total follicles in the ovarian cortex?

Thank you for this remark, which makes good sense to elaborate on. We checked the total follicle count (primordial, primary, secondary) once exposed to 10 μ M dorsomorphin or DMSO for all ovaries (n=16/17). The results showed that the ovaries cultured with DMSO contained 2207 ± 222.7 follicles, while the dorsomorphin-exposed ovaries contained 2229 ± 175.0 follicles. An unpaired t test showed that the differences were not significant (p=0.9391) and thus we can conclude that dorsomorphin does not change the number of total follicles but rather induces the activation of primordial follicles into primary or secondary follicles. This has been added to the results section to emphasize this result (Figure 2f).

3. Why the western blot in this manuscript are pretty faint? Is it because it represents only a few ovaries per lane?

That is a good point and yes, in most of the blots we pooled 4 ovaries, and therefore some of the blots might appear faint, and some might appear even more faint because of a colorimetric overlay with the marker. We have removed the overlay, which makes the blots slightly denser, and we have changed some of the membrane images to accommodate the journal request of not using images spliced together.

4. The AMPK role in follicle maturation is not clear. Specifically, in Figure 3, the author claims that Dorsomorphin treatment results in AMPK inhibition which results in more follicular maturation. And in Figure 1, the author also showed that primary follicles have lower AMPK expression, and possible lower AMPK activity. It is possible that Dorsomorphin induces follicles through an AMPK-independent mechanism, resulting in more maturation of follicles and, thus, less AMPK activity in the whole ovary. To delineate this problem, I suggest the author use other AMPK direct inhibitors, such as BAY-3827, and perform the same experiment as in Figure 2 and Figure 3. If the finding is

still the same as Dorsomorphin, then it is more likely that AMPK mediates the follicle maturation effect of Dorsomorphin.

It is true that there are few AMPK inhibitors available, and the best known is dorsomorphin. The inhibitory effect of dorsomorphin on AMPK has been confirmed by several others. We have added more explanation on AMPK inhibition in our revised manuscript and believe it adds more credibility to the use of dorsomorphin (Compound C).

5. I am confused with Western blot in Figure 4E and 4F, and related supplementary figures. Are these Western blot laser-captured follicles or the whole ovary? The reason that I am confused is that the author showed that Dorsomorphin increases follicle maturation, which means that if the whole ovary is used for Western blot, then we should see more FOXO3a and p27 in the cytosolic compartment from a higher fraction of matured follicles. If I understand the author's point correctly, the message of this figure is to show that Dorsomorphin activates primordial follicles in a PI3K/AKT-independent manner. In this case, the author might want to show that the FOXO3a cytosolic to nuclear ratio in the primordial and primary follicle is not statistically different between DMSO control and Dorsomorphin treated, instead of fractionated Western blot.

Thank you for pointing this out. It is a very important point to be absolutely clear about. All the Western blots are representing whole ovaries – it is not possible to do Western blotting on laser captured oocytes, and the laser-captured oocytes were only used for RNA sequencing.

The Western blotting data were used to measure changes in the levels of Foxo3a and p27 in whole ovaries. Then, we performed Western blotting on the cytoplasmic fractions, as you pointed out, to be the correct way to measure their activation by translocation. We observed that cytoplasmic levels of FOXO3a and p27 were increased in dorsomorphin-treated ovaries. In contrast, PTEN is decreased in the cytoplasm in dorsomorphin-treated ovaries. Jointly, this points towards primordial follicle activation.

Regarding the FOXO3A nuclear/cytoplasmic ratio, when a primordial follicle becomes primary, the protein moves from the nucleus to the cytoplasm and is then rapidly degraded by the secondary follicle stage (7) (also confirmed by IF staining). Therefore, as a larger amount of FOXO3A will be degraded and not just translocated, we did not find it relevant to explore the ratio.

We aim to demonstrate that AMPK controls follicular dormancy differently from PTEN-AKT signalling. Previous studies have shown that knocking down PTEN(8), FOXO3 null mice (9) and p27 (10) depleting the entire primordial follicle pool causes premature ovarian failure and retardation of oocyte growth. Currently, most work on the in vitro activation of primordial follicles has manipulated the PTEN pathway. Over time, it has been consistently observed that PTEN inhibition limits growth and survival rates, possibly due to increased DNA damage and reduced DNA repair in ovarian follicles, particularly in oocytes(11, 12). Therefore, it has been crucial for us to investigate whether exposure to dorsomorphin affects downstream regulators such as PTEN, FOXO3A, and p27, especially considering its potential clinical application.

Examining our follicle culture data, we also found that dorsomorphin exposure during primordial follicle activation does not adversely affect survival and MII rates. This may be attributed to the normal activity of PTEN, AKT, p27 and FOXO3A.

Minor,

1. Figure 1: The label is wrong; b should be moved to the heat map itself. The label 'd' is missing from the figure. Alternatively, include the heatmap description in Figure 1a description.

Thank you for pointing out this mistake. It has been corrected.

2. Line 112: Change figure reference from Figure 1b to Figure 2b.

This has been corrected.

3. Line 596: What is S2 reference to here? Does the author mean Figure S2?

We appreciate you bringing this error to our attention, and we have made the necessary correction.

4. Line 113 – 114: Change figure reference from Figure 1C to Figure 2C

This has been corrected.

5. Line 115: Change figure reference from Figure 1D to Figure 2D

This has been corrected.

6. Line 116: Change figure reference from Figure 1E to Figure 2E

This has been corrected.

7. Figure 4C: What is 'Tr', is the author pointing to granulosa cells? Can the author add this to the Figure description?

Thanks for this comment, this will of course be added to the figure description.

8. Line 675: in Figure S1 description, change FPRM to FPKM.

This has been corrected.

Reference

1. Lu, X., Guo, S., Cheng, Y., Kim, J., Feng, Y., & Feng, Y. (2017). Stimulation of ovarian follicle growth after AMPK inhibition. *Reproduction*, 153(5), 683-694. Retrieved Aug 10, 2023, from <https://doi.org/10.1530/REP-16-0577>

Reviewers' comments:

Reviewer #2 (Remarks to the Author):

As a key piece of evidence supporting the inhibition of primordial follicle activation by metformin (Fig. 2), the authors still chose to use a model of mouse ovarian culture. The ovary culture model is a very unstable model and the death and survival of primordial follicles per se are totally unpredictable. So the differences in follicle percentages seen by the authors are not reliable. If the authors want to show that metformin can really affect follicle activation, they can easily feed the postnatal mice with these medicine and show the in vivo, physiological follicle profiles, which are most reliable. In addition, as judged from the morphology in Fig. 2, b and d, metformin has no effect on the numbers of activated primordial follicles. Sections of the oval ovary from different directions would end up with the same set of photos.

The same problem goes with the culture of human ovarian tissue. First of all, the 49 year old tissue is nonsense as the age is close to menopause, so this sample should be excluded. Second of all, the follicle density and quantity varies tremendously in human ovarian cortex, therefore a sufficient number of repeats, for example, at least 3 samples from 3 patients who are at the reproductive age should be included.

Moreover, the presentation of percentages of follicle distribution is embarrassing, in a way that 1 primordial follicle and 2 primary follicles can be a totally different percentage for each type of follicles, from 2 primordial follicles and 1 primary follicle. It is well known that it is quite usual that a piece of human ovarian cortical tissue has 1 follicle or no follicle. Therefore, this reviewer is against the way that the authors present the follicle quantification data. The authors should present the absolute numbers of follicles at different stages in both the mouse ovary culture and human ovary tissue culture.

Reviewer #3 (Remarks to the Author):

I thank the author for addressing all my comments. In general, the manuscript is much improved from the previous version. Focusing more on the functional aspect of follicular function after Dorsomorphin-induced follicular dormancy is a good strategy and novel enough from other studies. I am satisfied with most of the responses that the author addressed. At this point, I would recommend a minor revision.

My only major concern is still the connection between AMPK and Primordial follicle enrichment. In the manuscript, the author claims that by inhibiting AMPK with Dorsomorphin, primordial follicles are preserved through an AMPK-dependent pathway. However, to make that mechanical claim, the author either has to orthogonally control AMPK activity through genetic manipulation OR use other AMPK inhibitor that is not Dorsomorphin and shows that Primordial Follicle is enriched in those conditions. My concern is not that Dorsomorphin is not inhibiting AMPK, but what else is Dorsomorphin perturbing that could lead to Primordial follicle enrichment found in this manuscript, regardless of AMPK inhibition.

Minor

1. Figure 1A, the heatmap color bar label is missing. I assume that it shows log fold change?
2. Figure 3B, the heatmap color bar label is also missing here.

Reply to Reviewers' comments:

Reviewer #2 (Remarks to the Author):

As a key piece of evidence supporting the inhibition of primordial follicle activation by metformin (Fig. 2), the authors still chose to use a model of mouse ovarian culture. The ovary culture model is a very unstable model and the death and survival of primordial follicles per se are totally unpredictable. So the differences in follicle percentages seen by the authors are not reliable. If the authors want to show that metformin can really affect follicle activation, they can easily feed the postnatal mice with these medicine and show the in vivo, physiological follicle profiles, which are most reliable. In addition, as judged from the morphology in Fig. 2, b and d, metformin has no effect on the numbers of activated primordial follicles. Sections of the oval ovary from different directions would end up with the same set of photos.

The same problem goes with the culture of human ovarian tissue. First of all, the 49 year old tissue is nonsense as the age is close to menopause, so this sample should be excluded. Second of all, the follicle density and quantity varies tremendously in human ovarian cortex, therefore a sufficient number of repeats, for example, at least 3 samples from 3 patients who are at the reproductive age should be included. Moreover, the presentation of percentages of follicle distribution is embarrassing, in a way that 1 primordial follicle and 2 primary follicles can be a totally different percentage for each type of follicles, from 2 primordial follicles and 1 primary follicle. It is well known that it is quite usual that a piece of human ovarian cortical tissue has 1 follicle or no follicle. Therefore, this reviewer is against the way that the authors present the follicle quantification data. The authors should present the absolute numbers of follicles at different stages in both the mouse ovary culture and human ovary tissue culture.

We employ gold-standard practices in our follicle culture assay. To begin with, our in vitro ovary culture system has been established by renowned follicle experts as documented in the study (doi:10.1016/j.fertnstert.2009.10.027). The methods employed are widely recognized and have been published in esteemed journals. The results are presented comprehensively, including both numerical values and percentages (e.g. <https://doi.org/10.1038/s41419-021-03842-1>, doi: 10.1096/fj.201900782R, <https://doi.org/10.1074/jbc.M113.532952>, <https://doi.org/10.1038/s41598-019-42878-4>, <https://doi.org/10.1002/ctm2.122>, DOI: 10.1186/s12915-018-0541-4, <https://doi.org/10.1038/s41419-021-03842-1>). As a proof of concept, numerical data have been incorporated with dorsomorphin, providing support for our validation. Additionally, numerical values are presented for our human ex vivo data. Obtaining patient tissue poses inherent challenges, and our data substantiate the proof of concept that the observed effect is translatable. There is considerable variation in the numbers of follicles among different patients and as follicles often clustered together, therefore within the same patient the variation can be considerable. Consequently, we find it more meaningful to present the data in percentage rather than absolute counts. To be included our analysis, a patient sample must have had a minimum of 10 counted follicles (will be stated in the discussion), as we are aware that cortical samples may exhibit a scarcity or absence of follicles. The human data primary serves to validate the effects identified in the mice. It is essential to note that we do not assert a significant difference in the human tissue; rather, we observe that these two treated patients appear to exhibit an activating effect similar to that seen in murine tissue.

Regarding the suggestion to include in vivo data on metformin, this has already been explored and documented in a relevant study (doi:10.1093/molehr/gaaa084). In that investigation, the authors

examined the impact of metformin in preventing the depletion of primordial follicles—an effect known to be induced by cyclophosphamide, causing severe gonadotoxicity through imbalanced activation of primordial follicles via PI3K/AKT/mTOR activation. Co-treatment with metformin successfully prevented the activation of primordial follicles, aligning with our in vitro observations. We have incorporated these findings into the discussion.

Reviewer #3 (Remarks to the Author):

I thank the author for addressing all my comments. In general, the manuscript is much improved from the previous version. Focusing more on the functional aspect of follicular function after Dorsomorphin-induced follicular dormancy is a good strategy and novel enough from other studies. I am satisfied with most of the responses that the author addressed. At this point, I would recommend a minor revision.

My only major concern is still the connection between AMPK and Primordial follicle enrichment. In the manuscript, the author claims that by inhibiting AMPK with Dorsomorphin, primordial follicles are preserved through an AMPK-dependent pathway. However, to make that mechanical claim, the author either has to orthogonally control AMPK activity through genetic manipulation OR use other AMPK inhibitor that is not Dorsomorphin and shows that Primordial Follicle is enriched in those conditions. My concern is not that Dorsomorphin is not inhibiting AMPK, but what else is Dorsomorphin perturbing that could lead to Primordial follicle enrichment found in this manuscript, regardless of AMPK inhibition.

Thank you very much for the acknowledgement of an improved revision. To address the remaining concern regarding the AMPK and primordial follicle enrichment, we adopted the suggested orthogonal approach to assess AMPK activity by employing the AMPK inhibitor BAY-3827 in our primary in vitro culture system. The data substantiate our observations and are depicted in Fig. S3. The results align with our dorsomorphin findings, demonstrating a stimulated activation of primordial follicles. We hope this settled the concern regarding the robust activation of primordial follicles upon AMPK inhibition.

Minor

1. Figure 1A, the heatmap color bar label is missing. I assume that it shows log fold change? FPKM values have been added.
2. Figure 3B, the heatmap color bar label is also missing here. Log2fold change will be added.

Reviewers' comments:

Reviewer #3 (Remarks to the Author):

I thank the author for addressing my comment. I am satisfied with the current state of the manuscript, and recommend for publication.

Reviewer #4 (Remarks to the Author):

In this study, the authors reported that an inhibited pathway of AMPK-WNT-FOXO3a may affect primordial follicle activation in conjunction with the PI3K/AKT signaling by applying mice models and human ovarian tissues. The authors also correlate these signalings to the high quality of mature oocytes. Despite that the topic of correlating AMPK pathway to the activation of primordial follicles in mice and human is meaningful, the data presented by the authors are not so much confident to convince me to agree with the conclusion. This is not only because the quality of some of the data are poorly provided, but because the manuscript is poorly written in highlighting the key novelty of the study. Maybe, the activation of primordial follicles have weak correlation with the quality of mature oocyte. Or, it is not suitable to directly conclude that successful activated primordial follicles deem to produce high quality mature oocytes. As is known, the development of the growing follicles may be affected by multiple factors during the long developmental progress. In the early and late stages of follicle growth, paracrine, autocrine and endocrine molecules, nutrients, blood supplying and other unidentified factors within the ovaries may affect the fate of the follicles. Only few follicles may have the potential to full develop and ovulate. Considering the inadequate solid data and not that much high standard figures, I do not think the manuscript is suitable to be published in the magazine. Some of the reasons are listed below.

1. In Fig. 2b, the enlarged image in the bottom row should indicate which region was selected from the above figure. The color of the hematoxylin stain in the enlarged image does not seem to match the above image.
2. The ovarian volume treated with 80 μ M metformin in Fig. S2c was significantly smaller than that in other groups, and most follicles in the ovarian cortex died. The figure should therefore be replaced.
3. Fig. 2a should indicate the number of days of mouse ovaries and the number of days of incubation after drug treatment.
4. The ovarian morphology of BAY-3827 group in Fig. S3b was abnormal and should be replaced.
5. In Fig. 2i, the ordinate expressions of the two statistical charts were inconsistent.
6. In Fig.4c and d, dsDNA should be changed to DAPI.
7. In Fig. 5a, the DAPI intensity between the control group and the treatment group was inconsistent, which seemed to be abnormal.
8. The manuscript is poorly written. I noticed that there are either long paragraphs in the introduction and discussion parts without specific key opinions of the authors, or short paragraphs lack of concise, such as in the last parts of discussion that seems to repeatedly give conclusions of the study but did not indicate the key point. Further, references of the manuscript are not uniformly organized in the revised version. For instances, the authors applied two kinds of format in line 316 through 324, page 14.

Reviewer #4 (Remarks to the Author):

In this study, the authors reported that an inhibited pathway of AMPK-WNT-FOXO3a may affect primordial follicle activation in conjunction with the PI3K/AKT signaling by applying mice models and human ovarian tissues. The authors also correlate these signalings to the high quality of mature oocytes. Despite that the topic of correlating AMPK pathway to the activation of primordial follicles in mice and human is meaningful, the data presented by the authors are not so much confident to convince me to agree with the conclusion. This is not only because the quality of some of the data are poorly provided, but because the manuscript is poorly written in highlighting the key novelty of the study. Maybe, the activation of primordial follicles have weak correlation with the quality of mature oocyte. Or, it is not suitable to directly conclude that successful activated primordial follicles deem to produce high quality mature oocytes. As is known, the development of the growing follicles may be affected by multiple factors during the long developmental progress. In the early and late stages of follicle growth, paracrine, autocrine and endocrine molecules, nutrients, blood supplying and other unidentified factors within the ovaries may affect the fate of the follicles. Only few follicles may have the potential to full develop and ovulate. Considering the inadequate solid data and not that much high standard figures, I do not think the manuscript is suitable to be published in the magazine. Some of the reasons are listed below.

We appreciate the reviewer for comments and for highlighting the relevance to mature oocytes. It was not our intention to imply that the activation of primordial follicles inherently yields quality oocytes. But having more primordial follicles activated, the likelihood of getting a better oocyte is increased. We have rephrased this section to accurately reflect this. We believe our data are properly conducted, with applied statistics and sufficient n number for our assay to reach significant results. We therefore are confident regarding the data interpretation and conclusion. We will address the remaining points raised by this reviewer below, point by point.

1. In Fig. 2b, the enlarged image in the bottom row should indicate which region was selected from the above figure. The color of the hematoxylin stain in the enlarged image does not seem to match the above image.

We thank the reviewer for pointing this out and agree. We indicated magnified regions from the ovary images, to match the selected area.

2. The ovarian volume treated with 80 μ M metformin in Fig. S2c was significantly smaller than that in other groups, and most follicles in the ovarian cortex died. The figure should therefore be replaced.

Yes, it was a smaller volume on that image. For each concentration, we took several images and picked one from our collection of images. We have substituted the ovary image with another image with a larger volume.

3. Fig. 2a should indicate the number of days of mouse ovaries and the number of days of incubation after drug treatment.

We have added this to the figure, to indicate number of days of mouse ovaries and number of incubation days with the drug.

4. The ovarian morphology of BAY-3827 group in Fig. S3b was abnormal and should be replaced.

As for comment 2, we take several images for each ovary and we have replaced the image in S3.

5. In Fig. 2i, the ordinate expressions of the two statistical charts were inconsistent.

The bar chart displays the distribution of follicles as percentages. This was done, given the substantial variability in follicle numbers among patients and the fact that follicles are often clustered within cortical pieces, we believe that visualizing the distribution as percentages is more appropriate for comparison. We have the total number of follicles and added this to Table 3, to provide this information for the percentage presentation in Fig 2i.

6. In Fig.4c and d, dsDNA should be changed to DAPI.

We have changed dsDNA to DAPI.

7. In Fig. 5a, the DAPI intensity between the control group and the treatment group was inconsistent, which seemed to be abnormal.

We have adjusted the DAPI intensity.

8. The manuscript is poorly written. I noticed that there are either long paragraphs in the introduction and discussion parts without specific key opinions of the authors, or short paragraphs lack of concise, such as in the last parts of discussion that seems to repeatedly give conclusions of the study but did not indicate the key point. Further, references of the manuscript are not uniformly organized in the revised version. For instances, the authors applied two kinds of format in line 316 through 324, page 14.

We greatly appreciate the feedback provided on the manuscript. However, we respectfully disagree with the assessment that it is poorly written. Prior to and following submission, we diligently utilized the Nature language editing service to ensure clarity and address any non-native English errors. Furthermore, we have thoroughly revised the manuscript to incorporate the reviewer's comments, particularly those pertaining to the potential increase in primordial follicles and oocyte quality. It is important to note that while increasing primordial follicle activation may lead to a greater number of high-quality eggs, this outcome cannot be guaranteed. Nevertheless, enhancing the activation of primordial follicles, with a focus on safety and balanced activation, holds promise for improving egg production. Additionally, it is crucial to acknowledge that follicle activation and growth are intricately linked to a complex interplay of autocrine, paracrine, and endocrine factors. We

have thoroughly addressed the comment and updated the text, checking for repetitive sections and ensuring consistency in formatting. Endnote has been employed to ensure seamless integration of references.